# Dissecting Representation Misalignment in Contrastive Learning via Influence Function

**Huanyi Xie**[1,2,*], **Chenyang Ren**[1,3,6,*], **Khouloud Saadi**[1,2], **Shu Yang**[1,2], **Zhen Tan**[4],
**Jingfeng Zhang**[5], **Lijie Hu**[3,†], **Di Wang**[1,2,†]

[1]King Abdullah University of Science and Technology (KAUST)
[2]Provable Responsible AI and Data Analytics (PRADA) Lab
[3]Mohamed bin Zayed University of Artificial Intelligence (MBZUAI)
[4]Arizona State University     [5]University of Auckland
[6]Shanghai Jiao Tong University (SJTU)

## Abstract

Contrastive learning, commonly applied in large-scale multimodal models, often relies on data from diverse and often unreliable sources, which can include misaligned or mislabeled text-image pairs. This frequently leads to robustness issues and hallucinations, ultimately causing performance degradation. Data valuation is an efficient way to detect and trace these misalignments. Nevertheless, existing methods are computationally expensive for large-scale models. Although computationally efficient, classical influence functions are inadequate for contrastive learning models, as they were initially designed for pointwise loss. Furthermore, contrastive learning involves minimizing the distance between positive sample modalities while maximizing the distance between negative sample modalities. This necessitates evaluating the influence of samples from both perspectives. To tackle these challenges, we introduce the Extended Influence Function for Contrastive Loss (ECIF), an influence function crafted for contrastive loss. ECIF considers both positive and negative samples and provides a closed-form approximation of contrastive learning models, eliminating the need for retraining. Building upon ECIF, we develop a series of algorithms for data evaluation, misalignment detection, and misprediction trace-back tasks. Experimental results demonstrate that our ECIF advances the transparency and interpretability of CLIP-style embedding models by offering a more accurate assessment of data impact and model alignment compared to traditional baseline methods.

## 1 Introduction

Contrastive learning has become a cornerstone in the development of multimodal models due to its ability to align representations of different modalities, such as images, text, and audio, within a shared semantic space (Chen et al., 2020b; Yin et al., 2023; Koh et al., 2024). However, models trained with contrastive learning often suffer from robustness issues (Radford et al., 2021) and hallucinations, which are mainly attributed to misaligned text-image pairs in training data (Kim et al., 2023). These misalignments, manifesting as semantic mismatches, contextual inconsistencies, or discrepancies between abstract and concrete elements, can severely degrade model performance. Contrastive learning relies on the assumption of consistent alignment between image-text pairs; however, when this assumption fails, it leads to incorrect interpretations, ultimately degrading model performance. Consequently, improving dataset transparency is crucial, as model developers need the ability to trace and identify problematic data samples. However, diagnosing issues caused by misaligned data, such as mislabeled or biased samples, is difficult when working with large text-image datasets.

---

[*]The first two authors contributed equally to this work.
[†]Correspondence to Lijie Hu (Email Address: `lijie.hu@mbzuai.ac.ae`) and Di Wang (Email Address: `di.wang@kaust.edu.sa`).

Although the critical role of training data in shaping the capabilities of multimodal models is well recognized, robust evaluation mechanisms for data quality remain lacking (Nguyen et al., 2022). To address this, various data valuation methods (Jia et al., 2019; Ghorbani & Zou, 2019; Yoon et al., 2020; Han et al., 2020) have been introduced to enhance dataset transparency by quantifying the contribution of individual data to model performance. These approaches typically assign higher contribution scores to training instances whose inclusion significantly boosts model performance compared to exclusion. Some methods, such as Shapley Value (Kwon & Zou, 2022), require multiple retraining processes with different data subsets, which is computationally expensive and impractical for large models. To overcome this limitation, influence function-based methods have gained popularity, as they estimate data contributions using gradient information, thereby avoiding retraining (Choe et al., 2024).

However, applying influence functions to models trained with contrastive learning presents significant challenges. (i) First, the influence function was initially designed for M-estimators (Huber, 1981), which operate with pointwise loss. However, multimodal models rely on noise-contrastive estimation (Radford et al., 2021; Gutmann & Hyvärinen, 2010; He et al., 2020) as their training objective. This objective encourages the model to draw positive pairs closer in feature space while pushing negative pairs apart, making the influence function unsuitable for direct application to contrastive loss. (ii) Second, the influence of negative pairs in contrastive learning has gained increasing attention recently (van den Oord et al., 2019; Yuksekgonul et al., 2023). Robinson et al. (2021) emphasized the importance of negative samples, especially "hard" negatives - samples that are mapped close in feature space but should ideally be far apart. However, the original definition of the influence function does not consider the roles of positive and negative samples. This oversimplified analysis is particularly prone to underestimating the impact of certain hard negative samples on the learning process (Chen et al., 2020a). (iii) Finally, computing the required gradients and Hessian matrices for influence functions is very demanding in terms of computational and memory resources, which makes it impractical in the large-scale high-dimensional environment of contrastive learning (Li et al., 2023a;b).

To address these challenges, we propose the *Extended Influence Function for Contrastive Loss (ECIF)*, a novel method designed to quantify data attribution specifically for contrastive learning. ECIF enjoys a closed-form approximation of the original contrastive loss, thus eliminating the need for re-training - a process that is impractical in the era of large models. It also accounts for the dual role of data points as both positive and negative samples, providing a more comprehensive understanding of their impact on model training. This approach provides a more accurate measurement of misalignment. Our contributions are summarized as follows:

- We propose ECIF, the first dual-perspective data valuation method for contrastive learning, which quantifies the impact of data points as both positive and negative samples. This comprehensive approach enables a more accurate measurement of data contribution, particularly evaluating the influence of negative samples in contrastive learning.

- Based on ECIF, we develop corresponding algorithms for different tasks, including identifying the most valuable data (related to specific tasks), misalignment detection, and misprediction trace-back.

- Comprehensive experimental results demonstrate that ECIF can effectively and efficiently remove the influence of samples compared to retraining and identify influential data in the training set. Moreover, our methods based on ECIF are also effective in identifying influential data (harmful data and valuable data) for fine-tuning, misprediction trace-back, and detecting misaligned data.

## 2 RELATED WORK

**Contrastive learning.** Recently, self-supervised contrastive learning (Chen et al., 2020c) has emerged as a highly effective approach to acquire representations without the need for labeled data (Donahue & Simonyan, 2019). This model utilizes a contrastive loss that pushes dissimilar data pairs apart while pulling similar pairs closer together. Contrastive learning plays an important role in the advancement of multimodal models by integrating and understanding information across diverse modalities, such as text and images (Radford et al., 2021; Jiang et al., 2024). In multimodal contrastive learning tasks, proper alignment of the training data ensures accurate cross-modal associations, enabling models to learn and extract consistent feature representations (Wang & Isola, 2020). One of the key challenges in training with noisy large-scale image-text pairs sourced from

the Internet is achieving effective alignment between these modalities. To address this, researchers have developed various methods, such as those proposed by Gao et al. (2022) and Yao et al. (2021), which introduce finer-grained and more extensive interactions between text and images to improve cross-modal alignment. Despite extensive research on contrastive learning, we are the first to explore interactive influence between pairs using influence functions. Our work bridges this gap by applying influence functions in contrastive learning, allowing for a deeper understanding of both positive and negative samples. This comprehensive approach improves the accuracy of misalignment measurements in data pairs, providing a more detailed assessment of the valuation of the data.

**Influence function.** Influence function, initially a staple in robust statistics (Cook, 2000; Cook & Weisberg, 1980), has seen extensive adoption within deep learning since Koh & Liang (2017). Its versatility spans various applications, including detecting mislabeled data, interpreting models, addressing model bias, and facilitating machine unlearning tasks. For data removal, recent work using influence function including unlearning features and labels (Warnecke et al., 2023), forgetting a subset of image data for training deep neural networks (Golatkar et al., 2020; 2021), removing the influence of nodes and edges in graph neural networks (Wu et al., 2023), and model debiasing (Chen et al., 2024). Besides, various studies have applied influence functions to interpret models across different domains, including natural language processing (Han et al., 2020) and image classification (Basu et al., 2021), while also addressing biases in classification models (Wang et al., 2019), word embeddings (Brunet et al., 2019), and finetuned models (Chen et al., 2020a). Recent advancements, such as the LiSSA method (Agarwal et al., 2017; Kwon et al., 2023; Grosse et al., 2023b) and kNN-based techniques (Guo et al., 2021), have been proposed to enhance the computational efficiency of computing the influence function. Despite numerous studies on influence functions, we are the first to extend them to contrastive learning. Moreover, compared to traditional models, contrastive learning introduces additional complexity in influence function analysis, as it requires considering data points in both positive and negative roles. Our dual-perspective approach of ECIF offers a comprehensive view of data impact, leading to accurate measurements of misalignment in text-image pairs. Bridging the theoretical gap between positive and negative pairs has posed significant challenges in our work, which has been addressed in our proof.

## 3 PRELIMINARIES

**Contrastive loss.** Contrastive loss is an effective tool in multimodal models for aligning and learning relationships between different types of data, such as images and text [1]. Specifically, given a set of paired data consisting of text $x^T$ and image $x^I$, we aim to construct embedding vectors $u$ and $v$ for text and image, respectively, via the encoder parameterized as $\theta$. In a batch of $N$ text-image pairs, each pair $(x_k^T, x_k^I)$ is embedded as $(u_k, v_k)$. We denote the text embeddings for this batch as $U = (u_1, \dots, u_N)$, and similarly, the image embeddings as $V = (v_1, \dots, v_N)$. Contrastive loss is designed to minimize the distance between embeddings of matching pairs while maximizing the distance between non-matching pairs. Define the cosine similarity function as $s(u, v) = \frac{u \cdot v^T}{\|u\|\|v\|}/\tau$, where $\tau$ is a trainable temperature parameter. For brevity, we will omit detailing $\tau$ in subsequent discussions. For each batch, we construct a similarity matrix $S$ with $S_{i,j} = s(u_i, v_j)$. Then, the self-supervised contrastive loss is defined as

$$L_{\text{Batch}}(U, V; \theta) \triangleq \sum_{i=1}^{N} -\log(e_i \cdot \sigma(S_{i,*}) - \log(e_i \cdot \sigma(S_{*,i}^{\mathrm{T}})), \tag{1}$$

where $e_i$ is the $i$-th standard basis vector in $N$-dimensional space, $\sigma$ is the softmax function. Observing from (1), we can separate the loss to image-to-text (I2T) and text-to-image (T2I) denoted as $L_{T2I}(u_i, V; \theta)$ and $L_{I2T}(v_i, U; \theta)$ and define loss function on similarity matrix as $L_{T2I}(S; \theta)$ (and $L_{I2T}(S; \theta)$). We will incorporate an $L_2$ regularization term into the loss function, which allows us to avoid overfitting. Thus, for a given set of batches $\mathcal{B}$, the objective loss can be written as $L_{\text{Total}}(\mathcal{B}; \theta) = \sum_{(U,V) \in \mathcal{B}} L_{\text{Batch}}(U, V; \theta) + \frac{\delta}{2}\|\theta\|_2^2$.

**Influence functions.** The influence function quantifies how an estimator relies on the value of each point in the sample. Consider a neural network $\hat{\theta} = \arg\min \sum_{i=1}^{n} \ell(z_i; \theta)$ with pointwise

---

[1]For simplicity, we focus on two modalities (text and image) in the paper. Our method can be generalized to multi-modalities directly.

loss function $\ell$ and dataset $D = \{z_i\}_{i=1}^n$. When we remove a point $z_m$ from the training dataset, the corresponding optimal model is denoted as $\hat{\theta}_{-z_m}$. The influence function provides an efficient way to approximate $\hat{\theta}_{-z_m} - \hat{\theta}$ for a strongly convex and twice differentiable $\ell$. By up-weighing $z_m$ by $\epsilon$, we denote the substitutional parameter via the $\epsilon$-parameterized response function as $\hat{\theta}_{-z_m}(\epsilon) = \arg\min \sum_{i=1}^n \ell(z_i; \theta) + \epsilon\ell(z_m; \theta)$. Then we can obtain an estimator for the actual change in parameters as:$\lim_{\epsilon\to-1} \hat{\theta}_{-z_m}(\epsilon) - \hat{\theta} = -H_{\hat{\theta}}^{-1} \cdot \nabla_\theta \ell(z_m; \hat{\theta})$, where $H_{\hat{\theta}} = \sum_{i=1}^n \nabla_\theta^2 \ell(z_i; \hat{\theta}) + \delta I$ is the Hessian matrix. For a differentiable model evaluation function $f$, such as calculating the total model loss over a test set, the change resulting from removing $z_m$ in the evaluation results can be approximated by $f(\hat{\theta}_{-z_m}) - f(\hat{\theta}) \approx \nabla_\theta f(\hat{\theta})(\hat{\theta}_{-z_m} - \hat{\theta}) \approx -\nabla_\theta f(\hat{\theta}) \cdot H_{\hat{\theta}}^{-1} \nabla_\theta \ell(z_m; \hat{\theta})$. Scaling gradient-based methods to contrastive loss is challenged by the high computational demands due to the gradients' high dimensionality. Choe et al. (2024) introduced a low-rank gradient projection algorithm (LOGRA) to enhance the efficiency of gradient projection. See Appendix C for more details.

## 4    INFLUENCE FUNCTION IN CONTRASTIVE LEARNING

In this section, we will consider how to estimate the attribution of a given sample $(x^T, x^I)$ to contrastive loss using the influence function method. Generally, in the original influence function method, the term in the loss function that only contains the complete information from the target sample is up-weighted by $\epsilon$. Then, a response function related to $\epsilon$ is derived. Within this analytical framework, when $\epsilon$ is set to $-1$, the resultant loss and model parameters are the same as those obtained by removing the sample via retraining. However, in the context of contrastive learning, because the information of the sample point appears in every term of the loss function for its batch, it is not feasible to isolate the relevant information of this sample within a batch into an independent term and then perform an up-weight operation on this sample to derive the influence function.

Thus, we must perform a fine-grained analysis of the specific contribution of sample $(x^T, x^I)$ within contrastive loss. Assume $(x^T, x^I)$ is assigned as the $n$-th pair in the $m$-th batch, in which the text and image data are embedded into matrix $U_m$ and $V_m$. Then $(x^T, x^I)$ serves as positive samples for each other in the $n$-th pairing loss $L_{T2I}(u_n, V_m; \theta)$ and $L_{I2T}(v_n, U_m; \theta)$ in (1). And it serves as a negative sample in other pairing losses. It can be seen through simple observation about (1) that when the data serves as a positive sample, its influence can be explicitly isolated. However, its information is coupled with other data when acting as a negative sample, necessitating further analysis. We provide the derivation of the influence function for these two scenarios separately.

### 4.1    INFLUENCE AS POSITIVE SAMPLES

To quantify the impact of $x^T$ and $x^I$ as positive samples, ideally, we can retrain the model after removing the corresponding $n$-th pairing tasks, i.e., removing $L_{T2I}(u_n, V_m; \theta)$ and $L_{I2T}(v_n, U_m; \theta)$ in the loss function. Thus, following the idea of influence function, we can up-weight these two parts by $\epsilon$ and obtain an up-weighted loss function as $L_{\text{Total},\epsilon}(\theta) = \sum_{(U,V)\in\mathcal{B}} L_{\text{Batch}}(U, V; \theta) + \frac{\delta}{2}\|\theta\|_2^2 + \epsilon \cdot \text{Pos}((x^T, x^I); \theta)$, with $\text{Pos}(x^T, x^I, \theta) = L_{T2I}(u_n, V_m; \theta) + L_{I2T}(v_n, U_m; \theta)$. And the parameters are obtained by $\hat{\theta}_\epsilon = \arg\min_\theta L_{\text{Total},\epsilon}(\theta)$. Then the influence function related to parameters can be deduced as:

$$\text{positive-IF}((x^T, x^I); \hat{\theta}) = -H_{\hat{\theta}}^{-1} \cdot \nabla_\theta \text{Pos}((x^T, x^I); \hat{\theta}). \tag{2}$$

where $H_{\hat{\theta}} = \nabla_\theta^2 \sum_{(U,V)\in\mathcal{B}} L_{\text{Batch}}(U, V; \hat{\theta}) + \delta I$ is the Hessian matrix. Proof refers to Appendix D.1.

**Extension to Multiple Samples.** The influence evaluation described above can be extended to a subset $\mathcal{D}^* \subset \mathcal{D}$. Let set $S$ to index the batches containing data from $\mathcal{D}^*$. For every $m \in S$, define an index set $E_m$ to specify the position of data from $\mathcal{D}^*$ within the $m$-th batch. We encapsulate the assigned results as $\text{Seg} = \{(m, E_m)|m \in S\}$. By employing a derivation method similar to that used for a single data point, we can obtain the parameter-related influence function for $\mathcal{D}^*$ by summing the influence (2) for all samples in $\mathcal{D}^*$.

**Proposition 4.1.** *The influence function for dataset $\mathcal{D}^*$ serving as positive samples (positive-IF) can be approximated by positive-IF$(\mathcal{D}^*, Seg; \hat{\theta}) = -H_{\hat{\theta}}^{-1} \cdot \nabla_\theta Pos(\mathcal{D}^*, Seg, \hat{\theta})$, where $Pos(\mathcal{D}^*, Seg; \hat{\theta}) = \sum_{m\in S} \sum_{n\in E_m} \left( L_{T2I}(u_n, V_m; \hat{\theta}) + L_{I2T}(v_n, U_m; \hat{\theta}) \right)$.*

## 4.2 INFLUENCE AS NEGATIVE SAMPLES

In Section 4.1, we quantified the impact of $x^T$ and $x^I$ as positive samples by removing related pairing tasks. Next, we estimate their impact as negative samples by removing them from tasks that serve as negative samples. To achieve this, we need to investigate the specific form of contrastive loss.

Taking the text2image (T2I) loss for the $k$-th text embedding $u_k$ as an example, we first calculate its similarity with all image embeddings in the batch to form a similarity vector $S(u_k, V)$, which is then processed through a softmax layer $\sigma(\cdot)$ to yield a probability distribution. The $k$-th element indicates the probability of correctly pairing the text $u_k$ with its corresponding image: $[\sigma(S(u_k, V))]_k = \frac{e^{S_{k,k}}}{\sum_{j \in [B]} e^{S_{k,j}}}$, where $B$ is the batchsize. The model is encouraged to enhance the probability of correct pairing by minimizing the negative logarithm of this value. For $n \neq k$, $v_n$ serves as a negative sample in this task and appears in the $S_{k,n}$ term in the denominator. Thus, after removing the impact of $(x^T, x^I)$ as a negative sample from the $m$-th batch, the loss function corresponding to this batch $L^m_{\text{T2I, -neg}}((x^T, x^I), S; \theta)$ should become:

$$\sum_{\substack{k \in [B] \\ k \neq n}} -\log \frac{e^{S_{k,k}}}{\sum_{\substack{j \in [B] \\ j \neq n}} e^{S_{k,j}}} + \text{Pos}((x^T, x^I); \theta). \tag{3}$$

The original influence function method evaluates a data point's impact by adjusting its weight via a separate term in the loss function and getting the response function. In contrastive learning, however, the influence of data points as negative samples is coupled with information from other data, which can observed from (3). We will separate an influence term related to the data effect when it serves as a negative sample. Actually, the modification in (3) is analogous to eliminating the $n$-th row and column from the original similarity matrix. Leveraging the idea of deriving the influence function, we aim to develop a response function that converges to the target loss by up-weighting some specific components.

Considering that similarities vectors are processed through the softmax layer, if we increase the similarity associated with $u_n$ and $v_n$ to a value approaching negative infinity, then after the exponential operation and the logarithmic function, the influence of $e^{S_{*,n}}$ and $e^{S_{n,*}}$ will become negligible. Mathematically, let $E_n$ be an $B \times B$ matrix such that its $n$-th column and the $n$-th row comprises ones, while all other entries are zero. We add the matrix $\log \zeta \times E_n$ to the similarity matrix. Then the loss function based on the revised similarity matrix becomes:

$$L^m_{\text{T2I},\zeta}((x^T, x^I), S; \theta) = \sum_{\substack{k \in [B] \\ k \neq n}} -\log \frac{e^{S_{k,k}}}{\sum_{j \in [B]} e^{S_{k,j}} + (\zeta - 1) \cdot e^{S_{k,n}}} + \text{Pos}((x^T, x^I); \theta). \tag{4}$$

We can easily see that as $\zeta$ approaches $0$, the loss function $L^m_{\text{T2I},\zeta}$ in (4) converges to $L^m_{\text{T2I, -neg}}$ in (3). When $\zeta = 1$, the loss function equals the original one. To further separate this influence as negative samples from the original loss function, we perform a Taylor expansion at $\zeta = 1$ and drop the $O((\zeta - 1)^2)$ term, then $L^m_{\text{T2I},\zeta}$ becomes

$$L^m_{\text{T2I}}(S; \theta) + (\zeta - 1) \cdot \sum_{\substack{k \in [B] \\ k \neq n}} \left( \frac{\sum_{j \in [B]} e^{S_{k,j}}}{e^{S_{k,n}}} \right) \xrightarrow{\zeta \to 0} L^m_{\text{T2I}}(S; \theta) - \sum_{\substack{k \in [B] \\ k \neq n}} \left( \frac{\sum_{j \in [B]} e^{S_{k,j}}}{e^{S_{k,n}}} \right),$$

and the left side is an estimation for (4). The minus term indicates the influence of $(x^T, x^I)$ as negative samples. By employing a similar method, one can obtain $L^m_{\text{I2T},\zeta}$ for the image2text part. Denote $\text{Neg}\left((x^T, x^I); \theta\right)$ as $\sum_{\substack{k \in [B] \\ k \neq n}} \left( \frac{\sum_{j \in [B]} e^{S_{k,j}}}{e^{S_{k,n}}} + \frac{\sum_{j \in [B]} e^{S_{j,k}}}{e^{S_{n,k}}} \right)$, Down-weighting the influence as a negative sample by $\zeta$ from 1 to 0, this influence in the loss function is then approximately eliminated. Then, the negative-influence function can be deduced as negative-IF$((x^T, x^I); \hat{\theta}) = -H_{\hat{\theta}}^{-1} \cdot \nabla_\theta \text{Neg}((x^T, x^I); \hat{\theta})$. As in the previous section, we can extend one single sample to a set of samples $\mathcal{D}^*$ with corresponding positional indices Seg.

**Proposition 4.2.** *The influence function for dataset $\mathcal{D}^*$ serving as negative samples (negative-IF) can be approximated by negative-IF$(\mathcal{D}^*, Seg; \hat{\theta}) = -H_{\hat{\theta}}^{-1} \cdot \nabla_\theta Neg(\mathcal{D}^*, Seg; \hat{\theta})$, with $Neg(\mathcal{D}^*, Seg; \hat{\theta}) \triangleq$*

$$\sum_{m \in S} \sum_{k \in [B]/E_m} \left( \frac{\sum_{j \in [B]} e^{S_{k,j}}}{\sum_{n \in E_m} e^{S_{k,n}}} + \frac{\sum_{j \in [B]} e^{S_{j,k}}}{\sum_{n \in E_m} e^{S_{n,k}}} \right).$$

Combining Proposition 4.1 and 4.2 together, we then define our influence function method on contrastive learning(ECIF) as follows.

**Definition 4.3** (ECIF). The extended influence function for contrastive loss (ECIF) of the target dataset $\mathcal{D}^*$ with its position index set $Seg = \{(m, E_m) | m \in S\}$ is defined as

$$\text{ECIF}(\mathcal{D}^*, \text{Seg}; \hat{\theta}) \triangleq \left( \text{positive-IF}(\mathcal{D}^*, \text{Seg}; \hat{\theta}), \text{negative-IF}(\mathcal{D}^*, \text{Seg}; \hat{\theta}) \right).$$

Thus, we can employ ECIF to estimate the changes in model parameters resulting from data removal.[2] We also give an upper bound on the error between the estimated influence given by ECIF and the actual influence obtained by model retraining in Appendix E for convex loss. We show that under certain scenarios, the approximation error becomes tolerable theoretically.

## 5 APPLICATIONS OF ECIF

We have proposed ECIF to evaluate the contribution of training data in contrastive learning. The ECIF method enables us to estimate the change in the learned parameters $\hat{\theta}$ if a training example pair is removed. Based on this, in this section, we will apply ECIF to two applications: misalignment detection and misprediction trace-back.

### 5.1 MISALIGNMENT DETECTION

Contrastive learning typically assumes a consistent alignment between all image-text pairs, and thus, misaligned data can lead to incorrect interpretations of these relationships, ultimately degrading model performance. Intuitively, given a high-quality validation data $D'$, if $D^*$ is a misaligned set, then the loss of $D'$ over the original model $\hat{\theta}$ should be greater than it over the model after deleting these misaligned data. And such a difference can be approximated by ECIF.

**Property 5.1.** *Considering a specific set $\mathcal{D}'$ with text and image embeddings $U'$ and $V'$, and a dataset $D^*$ to be removed, then we have*

$$L_{Batch}(U', V'; \hat{\theta}(-D^*)) - L_{Batch}(U', V'; \hat{\theta}) \approx \nabla L_{Batch}(U', V'; \hat{\theta})^{\mathrm{T}} (\hat{\theta}(-D^*) - \hat{\theta})$$

$$= -\nabla L_{Batch}(U', V'; \hat{\theta})^{\mathrm{T}} \cdot \left( posi\text{-}IF(\mathcal{D}^*, Seg; \hat{\theta}) + nega\text{-}IF(\mathcal{D}^*, Seg; \hat{\theta}) \right) \tag{5}$$

*where $\hat{\theta}(-D^*)$ is the optimal model for the loss eliminating $D^*$, posi-IF$(\mathcal{D}^*; Seg; \hat{\theta})$ and nega-IF$(\mathcal{D}^*, Seg; \hat{\theta})$ are obtained from Proposition 4.3 for $D^*$. We define term (5) as the task-related influence score, denoted as $IS(\mathcal{D}', \mathcal{D}^*, Seg; \hat{\theta})$.*

*Remark* 5.2. Task-related IS measures the impact of a data subset on task performance—its sign shows positive or negative influence, and its magnitude shows the extent. Thus, misalignment detection is formulated as $\arg\max_{\mathcal{D}^* \subset \mathcal{D}} IS(\mathcal{D}', \mathcal{D}^*, Seg; \hat{\theta})$ (see Algorithm 2 for details).

### 5.2 MISPREDICTION TRACE BACK

From a transparency perspective, when a model makes prediction errors on certain tasks, model trainers should be able to trace these error back to the specific training samples associated with the erroneous predictions. If we utilize the previous method for backtracking and choose the correct-labeled data that the model mispredicted to serve as the dataset $\mathcal{D}'$, then there is a significant possibility that the identified data are misaligned samples unrelated to the prediction errors.

This is because, in the definition of IS, the term on the right side of the multiplication sign quantifies the change in model parameters. Even if certain samples are unrelated to the task under analysis, they

---

[2]We abbreviate positive-IF as posi-IF and negative-IF as nega-IF.

may still exhibit a high IS due to their significant influence on model parameters. Therefore, compared to the application discussed above, it is necessary to constrain changes in the model parameters.

To address this, consider imposing a constraint $\rho$ on the permissible changes in model parameters when tracing back from mispredicted data while accounting for the process of upweighting the influence of samples as positive by $\epsilon$ and as negative by $\zeta$. Then we transform the trace back problem to identify which training example we should re-weight to most significantly impact the loss on the test sample set $\mathcal{D}'$ when given a small permissible change in model parameters.

$$\arg\max_{x \in \mathcal{D}, \epsilon, \zeta} |L_{\text{Batch}}(U', V'; \hat{\theta} + \Delta\hat{\theta}_{\epsilon,\zeta}(x)) - L_{\text{Batch}}(U', V'; \hat{\theta})| \quad \text{s.t.} \quad \left\|\Delta\hat{\theta}_{\epsilon,\zeta}(x)\right\|^2 \le \rho^2 \quad (6)$$

$$\approx \arg\max_{x \in \mathcal{D}, \epsilon, \zeta} |\nabla L_{\text{Batch}}(U', V'; \hat{\theta})^{\mathrm{T}} \Delta\hat{\theta}_{\epsilon,\zeta}(x)| \quad \text{s.t.} \left\|\Delta\hat{\theta}_{\epsilon,\zeta}(x)\right\|^2 \le \rho^2, \quad (7)$$

where $\Delta\hat{\theta}_{\epsilon,\zeta} = \epsilon \cdot \text{posi-IF}(x; \hat{\theta}) + (\zeta - 1) \cdot \text{nega-IF}(x; \hat{\theta})$ is the model parameter change estimated by ECIF when the influence of sample $x = (x^T, x^I)$ is upweighted by $\epsilon$ and $\zeta$.

**Proposition 5.3.** *Define* $I = [posi\text{-}IF(x; \hat{\theta}), nega\text{-}IF(x; \hat{\theta})]$. *If* $I^{\mathrm{T}} \cdot I$ *is irreversible, then Eq.(equation 7) is equivalent to* $\arg\max_{x \in \mathcal{D}} \|nega\text{-}IF(x; \hat{\theta})\|_2^{-1} |\nabla L_{Batch}(U', V'; \hat{\theta})^{\mathrm{T}} \cdot nega\text{-}IF(x; \hat{\theta})|$. *Else,* $I^{\mathrm{T}} \cdot I$ *is reversible, then (7) is equivalent to*

$$\arg\max_{x \in \mathcal{D}} \|\nabla L_{Batch}(U', V'; \hat{\theta})\|_2^{-1} \left|\nabla L_{Batch}(U', V'; \hat{\theta})^{\mathrm{T}} I \left[I^{\mathrm{T}} I\right]^{-1} I^{\mathrm{T}} \nabla L_{Batch}(U', V'; \hat{\theta})\right|.$$

The proposition above reduces the original argmax trace back problem to a simpler argmax problem. Consequently, we define the simplified argmax objective as a novel influence metric **relative-IS**. This metric, by adding constraints on parameter perturbations, helps us more accurately identify task-relevant samples. See Appendix Algorithm 4 for details.

# 6 EXPERIMENTS

Our ECIF method serves two main purposes: (1) approximating the performance of a retrained model without the need for actual retraining, and (2) enabling data attribution evaluation. In our experiments, we apply this method to various tasks, including removing data influence, identifying influential data (both harmful and valuable) for fine-tuning using task-related influence scores, tracing back mispredictions, and detecting misaligned data.

## 6.1 EXPERIMENTAL SETTINGS

**Datasets.** We will use the following datasets in our experiments: *FGVC-Aircraft dataset* (Maji et al., 2013), *Food101* (Bossard et al., 2014), *Flowers102* (Nilsback & Zisserman, 2008), *Cifar-10* (Krizhevsky et al., 2009), *Cifar100* (Krizhevsky et al., 2009), *Describable Textures Dataset (DTD)* (Sharan et al., 2014), and *Imagenette* (Deng et al., 2009) (a smaller subset of 10 classes from *Imagenet*).

**Our methods.** The experiments below directly implement the algorithms for the applications in the previous section. Algorithm 1 functions as the foundational algorithm, offering methods to calculate ECIF and providing a model editing based on ECIF. Algorithm 2 and 3 compute task-related IS in Prop. 5.1 to evaluate samples, indicating both the direction and intensity of their impact on the task. Meanwhile, Algorithm 4 is for relative-IS in Prop. 5.3, which aids in tracing back specific samples.

**Ground truth and baselines.** For all tasks, retraining from scratch will be ground truth, where the CLIP model is fine-tuned from scratch after sample removal. Furthermore, we compare ECIF with three other data attribution evaluation methods (IF-EKFAC (Grosse et al., 2023a), TARK (Park et al., 2023), and TracIN (Pruthi et al., 2020)) as baselines. Note that our ECIF is a direct implementation of Algorithm 1, utilizing positive and negative IF to modify the model for sample removal.

**Evaluation metric.** We utilize two main evaluation metrics to assess our models: accuracy and runtime (RT). Accuracy evaluates the model's performance by measuring the proportion of correctly

classified instances out of the total instances. Runtime,[3] measured in seconds, assesses the time required for each method to update the model. For implementation and additional details, please refer to Appendix G.1.

## 6.2 DATA INFLUENCE REMOVAL EVALUATION

We first consider the utility and efficiency of ECIF compared to ground truth. Table 2 shows that ECIF achieves a very close precision to retraining in all datasets, indicating that the models estimated by ECIF exhibit generalization capabilities similar to those obtained through retraining.

For example, the accuracy difference is only 0.30% on FGVCAircraft and 0.06% on Food101, which shows that ECIF effectively approximates the performance of retrained models. Moreover, ECIF is significantly faster, reducing runtimes by over 2 times compared to retraining (e.g., 456.0 seconds vs 1174.2 seconds on FGVCAircraft). These results highlight that ECIF can serve as a reliable surrogate for retraining, achieving comparable generalization performance. Moreover, ECIF can save approximately 80%-90% of the computational time required for retraining while maintaining comparable accuracy levels.

Table 2: Performance comparison of retraining and ECIF on different datasets.

| Sample | Method | FGVCAircraft | | Food101 | | Flowers102 | | Cifar100 | |
|---|---|---|---|---|---|---|---|---|---|
| | | Accuracy(%) | RT (s) | Accuracy(%) | RT (s) | Accuracy(%) | RT (s) | Accuracy(%) | RT (s) |
| Random | Retrain | 23.07±0.29 | 1174.2 | 84.93±0.17 | 875.4 | 68.16±0.22 | 995.4 | 73.50±0.35 | 753.6 |
| | ECIF | **22.77±0.09** | **456.0** | **84.87±0.24** | **436.8** | **68.53±0.12** | **437.4** | **73.00±0.20** | **444.0** |
| Valuable | Retrain | 22.93±0.33 | 933.6 | 84.80±0.16 | 952.8 | 68.23±0.33 | 985.8 | 73.50±0.41 | 721.2 |
| | ECIF | **22.73±0.09** | **357.0** | **84.86±0.05** | **376.2** | **68.26±0.12** | **391.2** | **72.92±0.31** | **462.0** |
| Harmful | Retrain | 23.50±0.11 | 1344.0 | 84.83±0.05 | 875.4 | 68.00±0.16 | 965.4 | 72.83±0.12 | 775.2 |
| | ECIF | **23.02±0.07** | **375.6** | **84.90±0.01** | **373.2** | **68.30±0.01** | **376.2** | **73.00±0.20** | **435.6** |

We then compare the effectiveness of ECIF with previous baseline methods in assessing data attribution. We begin by using various methods to perform a data attribution evaluation. Subsequently, we remove 10% of the most harmful samples from the training dataset and retrain the model. Specifically, for ECIF, we retrain the model and utilize it to update the results accordingly. The results are presented in Table 1. We can see that when harmful data identified

Table 1: Performance comparison with baselines.

| Method | Accuracy(%) | | | |
|---|---|---|---|---|
| | FGVCAircraft | Food101 | Flowers102 | Cifar100 |
| **Original model without data removal** | | | | |
| **Fine-tune** | 22.18±0.21 | 83.85±0.12 | 67.64±0.31 | 72.31±0.11 |
| **Harmful samples ↑** | | | | |
| Retrain | 23.50±0.11 | 84.83±0.05 | 68.00±0.16 | 72.83±0.12 |
| IF-EKFAC | 19.84±0.08 | 78.26±0.09 | 60.74±0.23 | 61.67±2.49 |
| TRAK | 18.27±0.09 | 77.27±0.03 | 59.21±0.02 | 58.67±1.70 |
| TracIN | 19.48±0.10 | 78.35±0.01 | 60.60±0.03 | 59.00±6.16 |
| **ECIF(ECIF)** | **23.02±0.07** | **84.90±0.01** | **68.30±0.01** | **73.00±0.20** |

by ECIF are removed, the accuracy of both the retrained model and ECIF's edited version aligns closely and both are higher than those observed with no data removal. This suggests that ECIF is capable not only of accurately editing the model but also of effectively identifying influential data. We can also see that the baseline methods have a significant gap with ECIF or Retrain, indicating that they are less efficient. This is because these methods are tailored for pointwise loss and are inapplicable to contrastive loss. In addition, we observe that all previous methods fail to identify harmful samples. This is because previous methods focus mainly on the most valuable/influential data, while for each sample there is both positive and negative influence in contrastive learning. We also study performance across varying numbers of removal samples (see Appendix G.3).

## 6.3 IDENTIFYING INFLUENTIAL DATA FOR FINE-TUNING

**Task-related IS can identify valuable data.** To numerically assess the precision of data valuation algorithms, we employ the brittleness test (Ilyas et al., 2022), which evaluates the algorithm's ability to accurately identify the most valuable data for a specific task. Our evaluation process is as follows: utilizing the validation set within Algorithm 2, we compute the task-related IS for each individual training data point. We then remove the top-$k$ valuable data points, with $k$ ranging from 5% to 30%, retrain the model multiple times using different random seeds, and assess the resultant change in overall model accuracy. The results in Figure 1b reveal that the removal of valuable data identified by ECIF leads to a consistent decrease in the accuracy of the model, from 84.7 to 84.1. In contrast, the removal of random data triggers an increase in model accuracy once the removal proportion reaches

---

[3]Here, runtime refers to the total runtime required for the updating process.

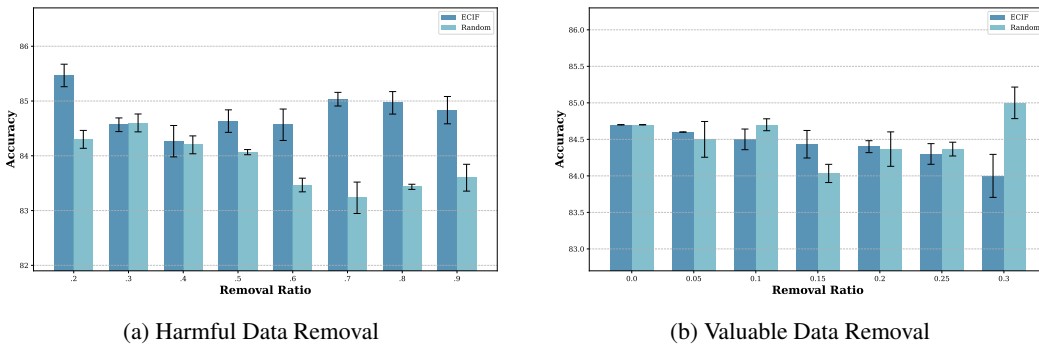

(a) Harmful Data Removal

(b) Valuable Data Removal

Figure 1: Accuracy after data removal on Food101.

0.3. This suggests that Food101 contains substantial noise and that our algorithm can effectively identify data points that genuinely enhance the predictive accuracy of the model.

**Task-related IS can identify harmful data.** Algorithm 2 identifies data pairs with negative task-related IS as harmful data for the task. To demonstrate the effectiveness of our algorithm in identifying detrimental data for specific tasks, we conducted experiments on several noisy datasets, such as Food101. See Appendix G.2 for the results of other noisy data.

We collected the harmful data identified by ECIF and then retrained the model multiple times with varying harmful data removal ratios and different random seeds. We compared its accuracy to that of a model retrained after randomly removing the same number of data points. The results in Figure 1a demonstrate the effectiveness of our approach in improving the performance of the model by eliminating harmful data using IS related to tasks. Figure 1a indicates that with varying proportions of harmful data removal, the accuracy of the retrained model consistently fluctuates around its original level. When $10\%$ of harmful data is removed, accuracy increases by approximately $1\%$. In contrast, with random deletions, the accuracy continues to decrease. This suggests that the accuracy improvement from the removal of harmful data with ECIF is not merely due to the removal action itself, but rather because the removed data genuinely had a detrimental effect on model training.

## 6.4 VISUALIZATION OF MISPREDICTION TRACE BACK

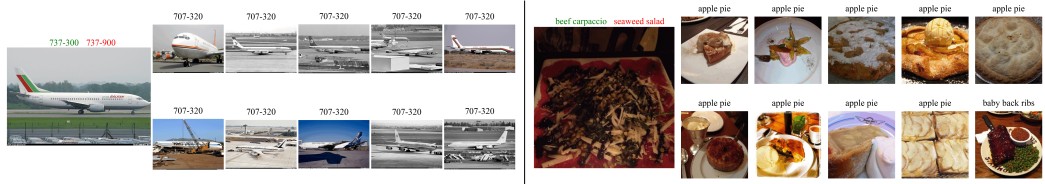

Table 3: Top-10 related training data traced by mispredicted data.

We apply Algorithm 4 to identify training data that are most relevant to specific mispredicted test samples. In this process, we select samples in the test data on which the model made a misclassification. Using the relative IS, we can identify the training data with the highest influence on the misprediction and visualize it. Table 3 shows the results of this misprediction trace-back process (see Appendix G.4 for additional results). Each pair of images compares a test sample with its most influential training counterpart. On the left, we show examples from the test set where the model produced incorrect predictions. On the right, the corresponding training data are shown, i.e., these data points hold the highest relative ISs in relation to the mispredicted test samples. This comparison helps shed light on how specific training samples may have contributed to the incorrect output of the model. According to the visualization results, it can be observed that the samples traced back to the original task exhibit similarities in shape or texture with the original task.

## 6.5 DATASET CLEANING: MISALIGNMENT DETECTION

We employ relative IF to detect misaligned pairs. Regarding the selection of validation dataset, we experiment with two approaches: randomly selecting from the gold dataset (Algorithm 2) and calculating based on the influence of the evaluated sample points (Algorithm 3), in which the test loss

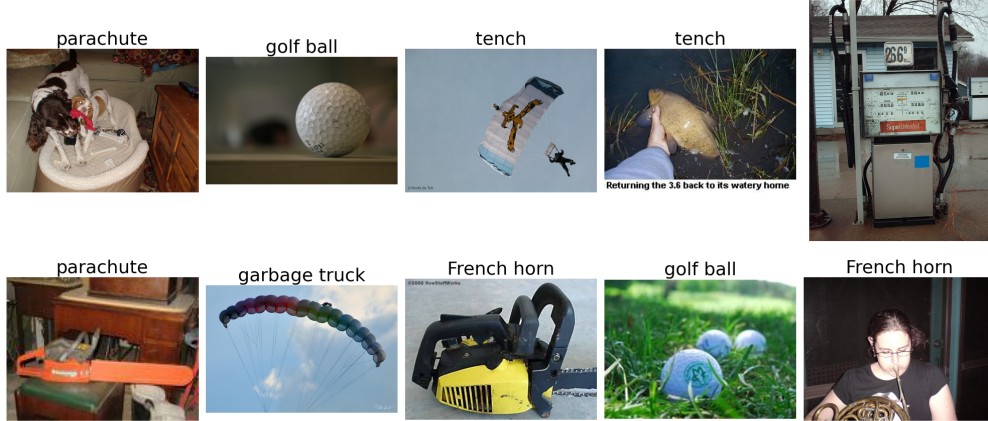

Figure 2: Top-10 misaligned pairs in the 20% mislabeled dataset.

is defined as CLIP score (Hessel et al., 2021) of the evaluated data pair. We first mislabel 10%-30% training samples and then identify the misaligned pairs by selecting those with the highest negative IS. These pairs are visualized in Figure 2 (see Appendix G.5 for additional results). The visualization results reveal that 8 of the top 10 data points with the highest IS are entirely contained within the mislabeled portion of our training dataset. This suggests that our algorithm has effectively identified the noise data artificially introduced into the dataset.

## 7 CONCLUSION

We propose ECIF (Extended Influence Function for Contrastive Loss), a novel data valuation framework for contrastive learning. ECIF evaluates both positive and negative sample impacts through dual-perspective analysis, enabled by efficient closed-form approximation without retraining. Validated in real-world applications, ECIF effectively supports fine-tuning, misprediction tracing, and misalignment detection while maintaining scalability for large models.

## ACKNOWLEDGMENT

Di Wang, Huanyi Xie, Chenyang Ren, Khouloud Saadi, and Shu Yang are supported in part by the funding BAS/1/1689-01-01,RGC/3/7125-01-01, FCC/1/5940-20-05, FCC/1/5940-06-02, and King Abdullah University of Science and Technology (KAUST) – Center of Excellence for Generative AI, under award number 5940 and a gift from Google. Lijie Hu is supported by the funding BF0100 from Mohamed bin Zayed University of Artificial Intelligence (MBZUAI). For computer time, this research used lbex managed by the Supercomputing Core Laboratory at King Abdullah University of Science & Technology (KAUST) in Thuwal, Saudi Arabia.

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

## A    THE USE OF LARGE LANGUAGE MODELS (LLMS)

We used LLMs to refine grammar and improve language fluency. The authors reviewed and edited all LLM-generated content and assume full responsibility for the final text.

## B    ALGORITHM

---

**Algorithm 1** ECIF

---

1: **Input:** Training Dataset $\mathcal{D} = \{(x^T, x^I)\}$, dataset $\mathcal{D}^*$ to be evaluated, the parameters $\hat{\theta}$ which is involved in IF calculation in the model and the regularization parameter $\delta$.
2: Define $S(\cdot, \cdot)$ as the similarity score.
3: Compute the text embedding and image embedding for $\mathcal{D}^*$ as uu and vv.
4: Random divide the training dataset $\mathcal{D}$ into MM batches and obtain the position index of $\mathcal{D}^*$ as $\mathrm{Seg} \triangleq \{(m, E_m)|m \in S\}$.
5: Compute the influence term as positive and negative samples for the mm-th batch in $S$ by:

$$\mathrm{Pos}_m = \sum_{n \in E_m} \left( -\log \frac{e^{S(u_n, v_n)}}{\sum_{j=1}^N e^{S(u_n, v_j)}} - \log \frac{e^{S(v_n, u_n)}}{\sum_{j=1}^N e^{S(v_n, u_j)}} \right)$$

$$\mathrm{Neg}_m = \sum_{i \in [N]/E_m} \left( \frac{\sum_{j=1}^N e^{S(u_i, v_j)}}{\sum_{n \in E_m} e^{S(u_i, v_n)}} + \frac{\sum_{j=1}^N e^{S(u_i, v_j)}}{\sum_{n \in E_m} e^{S(v_i, u_n)}} \right)$$

6: Compute the sum of the gradient of $\mathrm{Pos}_m$ and $\mathrm{Neg}_m$ as

$$\tilde{\mathrm{Pos}} = \sum_{m \in S} \nabla_\theta \mathrm{Pos}_m, \text{ and } \tilde{\mathrm{Neg}} = \sum_{m \in S} \nabla_\theta \mathrm{Neg}_m.$$

7: Compute the batch embedding for $\mathcal{D}$ as $\{B_m, m \in [M]\}$.
8: Compute the inverse Hessian matrix of the loss function with respect to $\hat{\theta}$ as

$$G = \left[ \sum_{m \in [M]} \nabla_\theta^2 L_{\mathrm{Batch}}(B_m; \hat{\theta}) + \delta \cdot I \right]^{-1}$$

9: Compute the positive-IF$(\mathcal{D}^*, \mathrm{Seg})$ and negative-IF$(\mathcal{D}^*, \mathrm{Seg})$ as:

$$\text{positive-IF}(\mathcal{D}^*, \mathrm{Seg}) = -G \cdot \tilde{\mathrm{Pos}}, \quad \text{negative-IF}(\mathcal{D}^*, \mathrm{Seg}) = -G \cdot \tilde{\mathrm{Neg}}$$

10: Obtain the ECIF as

$$\mathrm{ECIF}(\mathcal{D}^*, \mathcal{D}) = (\text{positive-IF}, \text{negative-IF}) \tag{8}$$

11: Edit model parameter to unlearn dataset $\mathcal{D}^*$ by

$$\tilde{\theta} = \hat{\theta} - \text{positive-IF} - \text{negative-IF}$$

12: **Return:** ECIF$(\mathcal{D}^*, \mathcal{D})$, Edited parameter $\tilde{\theta}$.

---

## C    ACCELERATION FOR INFLUENCE FUNCTION

**LOGRA.**    They observed that the gradient from backpropagation is structured as a sum of Kronecker products of forward and backward activations. LOGRA applies an additional Kronecker-product

---

**Algorithm 2** Task-related Influence Score Based on ECIF

---

1: **Input:** Training Dataset $\mathcal{D} = \{(x^T, x^I)\}$, dataset $\mathcal{D}^*$ to be evaluated, test dataset $\mathcal{D}'$, the parameters $\hat{\theta}$ which is involved in IF calculation in the model.
2: Compute the $\mathrm{ECIF}(\mathcal{D}^*, \mathcal{D}) = (\text{positive-IF}, \text{negative-IF})$ by algorithm 1.
3: Compute the gradient of the batch loss function of the test data as

$$C = \sum_{(U,V) \in \mathcal{D}'} \nabla_\theta L_{\mathrm{Batch}}(U, V; \hat{\theta})$$

4: Compute the task-related influence score as

$$IS = C^{\mathrm{T}} \cdot \text{positive-IF} + C^{\mathrm{T}} \cdot \text{negative-IF}$$

5: **Return:** Task-related Influence Score $IS$.

---

**Algorithm 3** Self Influence Score Based on ECIF

---

1: **Input:** Training Dataset $\mathcal{D} = \{(x^T, x^I)\}$, dataset $\mathcal{D}^*$ to be evaluated, test dataset $\mathcal{D}'$, the parameters $\hat{\theta}$ which is involved in IF calculation in the model.
2: Compute the $\mathrm{ECIF}(\mathcal{D}^*, \mathcal{D}) = (\text{positive-IF}, \text{negative-IF})$ by algorithm 1.
3: Compute the gradient of the batch loss function of the test data as

$$C = \frac{1}{|\mathcal{D}'|} \sum_{(x^T, x^I) \in \mathcal{D}'} \nabla_\theta - \log \frac{u^{\mathrm{T}} \cdot v}{\|u\| \cdot \|v\|},$$

where $u$ and $v$ is the embedding for $x^T$ and $x^I$
4: Compute the task-related influence score as

$$IS = C^{\mathrm{T}} \cdot \text{positive-IF} + C^{\mathrm{T}} \cdot \text{negative-IF}$$

5: **Return:** Task-related Influence Score $IS$.

---

**Algorithm 4** Relative Influence Score Based on ECIF

---

1: **Input:** Training Dataset $\mathcal{D} = \{(x^T, x^I)\}$, dataset $\mathcal{D}^*$ to be evaluated, test dataset $\mathcal{D}'$.
2: Compute the $\mathrm{ECIF}(\mathcal{D}^*, \mathcal{D}) = (\text{positive-IF}, \text{negative-IF})$ by algorithm 1.
3: Compute the gradient of the batch loss function of the test data as

$$C = \sum_{(U,V) \in \mathcal{D}'} \nabla_\theta L_{\mathrm{Batch}}(U, V; \hat{\theta})$$

4: **if** positive-IF is parallel to negative-IF **then**
5:    Compute the relative-IS as

$$\text{relative-IS} = \|\text{positive-IF}\|^{-1} \left| C^{\mathrm{T}} \text{negative-IF} \right|$$

6: **else** {positive-IF is not parallel to negative-IF}
7:    Define $I = [\text{positive-IF}, \text{negative-IF}]$.
8:    Compute the relative-IS as

$$\text{relative-IS} = \|C\|^{-1} \left| C^{\mathrm{T}} I \left[ I^{\mathrm{T}} \cdot I \right]^{-1} I^{\mathrm{T}} C \right|$$

9: **end if**
10: **Return:** relative-IS.

---

---

**Algorithm 5** Relative Influence Score Based on ECIF

---

1: **Input:** Training Dataset $\mathcal{D} = \{(x^T, x^I)\}$, dataset $\mathcal{D}^*$ to be evaluated, test dataset $\mathcal{D}'$.
2: Compute the $\text{ECIF}(\mathcal{D}^*, \mathcal{D}) = (\text{positive-IF}, \text{negative-IF})$ by algorithm 1.
3: Compute the gradient of the batch loss function of the test data as

$$C = \frac{1}{|\mathcal{D}'|} \sum_{(x^T, x^I) \in \mathcal{D}'} \nabla_\theta - \log \frac{u^T \cdot v}{\|u\| \cdot \|v\|},$$

where $u$ and $v$ is the embedding for $x^T$ and $x^I$
4: **if** positive-IF is parallel to negative-IF **then**
5:     Compute the relative-IS as

$$\text{relative-IS} = \|\text{positive-IF}\|^{-1} \left| C^T \text{negative-IF} \right|$$

6: **else** {positive-IF is not parallel to negative-IF}
7:     Define $I = [\text{positive-IF}, \text{negative-IF}]$.
8:     Compute the relative-IS as

$$\text{relative-IS} = \|C\|^{-1} \left| C^T I \left[ I^T \cdot I \right]^{-1} I^T C \right|$$

9: **end if**
10: **Return:** relative-IS.

---

structure to the projection matrix $P \triangleq P_i \otimes P_o$. It first projects the forward and backward activations onto low-dimensional spaces using $P_i$ and $P_o$, respectively, and then reconstructs the projected gradient directly from these reduced activations. For one layer, given the input $x_i$, output $x_o$ and the weight $W$, the forward and backward computation can be written as $x_o = W x_i$, $\text{vec}(\mathcal{D}W) = \sum_{t=1}^{T} x_{i,t} \otimes \mathcal{D}x_{o,t}$, $\mathcal{D}x_i = W^T \mathcal{D}x_o$, where $T$ denotes for the sequence dimension in language modeling, $\mathcal{D}$ the derivative with respect to the loss, $\otimes$ the Kronecker product, and $\text{vec}(\cdot)$ the vectorization operation. Observing gradient $\text{vec}(\mathcal{D}W)$ obtained during backpropagation is structured as a sum of Kronecker products between forward and backward activations, LOGRA imposes an additional Kronecker-product structure on the projection matrix P as follows:

$$P\text{vec}(\mathcal{D}W) \triangleq (P_i \otimes P_o)\text{vec}(\mathcal{D}W) = \sum_{t=1}^{T} (P_i \otimes P_o)(x_{i,t} \otimes \mathcal{D}x_{o,t}) = \sum_{t=1}^{T} P_i x_{i,t} \otimes P_o \mathcal{D}x_{o,t}$$

where $P_i$ is the projection matrix on the input and $P_o$ is that on the backward activations, and $P \triangleq P_i \otimes P_o$.

## D    INFLUENCE FUNCTION IN CONTRASTIVE LEARNING

### D.1    INFLUENCE FUNCTION FOR POSITIVE SAMPLES.

We first consider the influence function for positive samples.

**Single Data Pair Version.**    To quantify the impact of $x^T$ and $x^I$ as positive samples, we first define $L_{T2I}(u_n, V_m; \theta) + L_{I2T}(v_n, U_m; \theta)$ as $\text{Pos}((x^T, x^I); \theta)$. Following the idea of influence function, we can up-weight these two parts by $\epsilon$ and obtain an up-weighted loss function as

$$L_{\text{Total},\epsilon}(\theta) = \sum_{(U,V) \in \mathcal{B}} L_{\text{Batch}}(U, V; \theta) + \frac{\delta}{2}\|\theta\|_2^2 + \epsilon \cdot \text{Pos}((x^T, x^I); \theta).$$

And the parameters are obtained by $\hat{\theta}_\epsilon = \arg\min_\theta L_{\text{Total},\epsilon}(\theta)$. From this minimizing condition, we have

$$\sum_{(U,V) \in \mathcal{B}} \nabla_\theta L_{\text{Batch}}(U, V; \hat{\theta}_\epsilon) + \epsilon \cdot \nabla_\theta \text{Pos}((x^T, x^I); \hat{\theta}_\epsilon) = 0$$

Perform a Taylor expand at $\theta = \hat{\theta}$, we have

$$\sum_{(U,V)\in\mathcal{B}} \nabla_\theta L_{\text{Batch}}(U,V;\hat{\theta}) + \epsilon \cdot \nabla_\theta \text{Pos}((x^T,x^I);\hat{\theta}) + \sum_{(U,V)\in\mathcal{B}} \nabla_\theta^2 L_{\text{Batch}}(U,V;\hat{\theta}) \cdot \left(\hat{\theta}_\epsilon - \hat{\theta}\right) \approx 0$$

Because $\hat{\theta}$ minimizes $\sum_{(U,V)\in\mathcal{B}} L_{\text{Batch}}(U,V;\hat{\theta})$, the first term in the above equation equals 0.

$$\text{positive-IF}((x^T,x^I);\hat{\theta}) \triangleq \left.\frac{d\hat{\theta}_\epsilon}{d\epsilon}\right|_{\epsilon=0} = -H_{\hat{\theta}}^{-1} \cdot \nabla_\theta \text{Pos}((x^T,x^I);\hat{\theta})$$

where $H_{\hat{\theta}} = \nabla_\theta^2 \sum_{(U,V)\in\mathcal{B}} L_{\text{Batch}}(U,V;\hat{\theta}) + \delta I$ is the Hessian matrix at $\hat{\theta}$.

**Extension to Multiple Samples.** The influence evaluation described above can be extended to a subset $\mathcal{D}^* \subset \mathcal{D}$. Let set $S$ to index the batches containing data from $\mathcal{D}^*$. For every $m \in S$, define an index set $E_m$ to specify the position of data from $\mathcal{D}^*$ within the $m$-th batch. We encapsulate the assigned results as $\text{Seg} = \{(m, E_m)|m \in S\}$. By employing a derivation method similar to that used for a single data point, we can obtain the parameter-related influence function for $\mathcal{D}^*$ by summing the influence as a position sample (2) for all samples in $\mathcal{D}^*$.

**Proposition D.1.** *The influence function for dataset $\mathcal{D}^*$ serving as positive sample (positive-IF) is*

$$positive\text{-}IF(\mathcal{D}^*, Seg; \hat{\theta}) = -H_{\hat{\theta}}^{-1} \cdot \nabla_\theta Pos(\mathcal{D}^*, Seg; \hat{\theta})$$

*where*

$$Pos(\mathcal{D}^*, Seg; \hat{\theta}) = \sum_{m\in S} \sum_{n\in E_m} \left( L_{T2I}(u_n, V_m; \hat{\theta}) + L_{I2T}(v_n, U_m; \hat{\theta}) \right)$$

*Proof.* $\text{Seg} = \{(m, E_m)|m \in S\}$, for $m \in S$, $U_m$, $V_m$ are the text and image embedding for the $m$-th batch, respectively. For $n \in E_m$, $u_n$ and $v_n$ are embeddings for a single data pair in $m$-th batch, which is included in the dataset to be evaluated $\mathcal{D}^*$. Define $\text{Pos}(\mathcal{D}^*, \text{Seg}; \theta)$ as

$$\text{Pos}(\mathcal{D}^*, \text{Seg}; \theta) = \sum_{m\in S} \sum_{n\in E_m} \left( L_{T2I}(u_n, V_m; \theta) + L_{I2T}(v_n, U_m; \theta) \right)$$

Following the idea of influence function, we can up-weight these by $\epsilon$ and obtain an up-weighted loss function as

$$L_{\text{Total},\epsilon}(\theta) = \sum_{(U,V)\in\mathcal{B}} L_{\text{Batch}}(U,V;\theta) + \frac{\delta}{2}\|\theta\|_2^2 + \epsilon \cdot \text{Pos}(\mathcal{D}^*, \text{Seg}; \theta).$$

And the parameters are obtained by $\hat{\theta}_\epsilon = \arg\min_\theta L_{\text{Total},\epsilon}(\theta)$. From this minimizing condition, we have

$$\sum_{(U,V)\in\mathcal{B}} \nabla_\theta L_{\text{Batch}}(U,V;\hat{\theta}_\epsilon) + \epsilon \cdot \nabla_\theta \text{Pos}(\mathcal{D}^*, \text{Seg}; \hat{\theta}_\epsilon) = 0$$

Perform a Taylor expand at $\theta = \hat{\theta}$, we have

$$\sum_{(U,V)\in\mathcal{B}} \nabla_\theta L_{\text{Batch}}(U,V;\hat{\theta}) + \epsilon \cdot \nabla_\theta \text{Pos}(\mathcal{D}^*, \text{Seg}; \hat{\theta}) + \sum_{(U,V)\in\mathcal{B}} \nabla_\theta^2 L_{\text{Batch}}(U,V;\hat{\theta}) \cdot \left(\hat{\theta}_\epsilon - \hat{\theta}\right) \approx 0$$

Because $\hat{\theta}$ minimizes $\sum_{(U,V)\in\mathcal{B}} L_{\text{Batch}}(U,V;\hat{\theta})$, the first term in the above equation equals 0.

$$\text{positive-IF}(\mathcal{D}^*, \text{Seg}; \hat{\theta}) \triangleq \left.\frac{d\hat{\theta}_\epsilon}{d\epsilon}\right|_{\epsilon=0} = -H_{\hat{\theta}}^{-1} \cdot \nabla_\theta \text{Pos}(\mathcal{D}^*, \text{Seg}; \hat{\theta})$$

where $H_{\hat{\theta}} = \nabla_\theta^2 \sum_{(U,V)\in\mathcal{B}} L_{\text{Batch}}(U,V;\hat{\theta}) + \delta I$ is the Hessian matrix at $\hat{\theta}$. $\square$

## D.2 INFLUENCE FUNCTION FOR NEGATIVE SAMPLES.

Then, we come to derive the influence function for the negative sample.

In this part, we will illustrate how we give an approximation function for the loss function in which the influence as a negative sample of the data we are considering is removed. With the help of Taylor expansion, this influence is separated into a single term in this approximation function, and we can achieve this by removing this term from the original loss function.

After removing the impact of $(x^T, x^I)$ as a negative sample from the $m$-th batch, the loss function corresponding to this batch should become:

$$L_{\text{T2I,-neg}}^m((x^T, x^I), S; \theta) = \sum_{\substack{k \in [B] \\ k \neq n}} \frac{e^{S_{k,k}}}{\sum_{\substack{j \in [B] \\ j \neq n}} e^{S_{k,j}}} + \text{Pos}((x^T, x^I); \theta). \tag{9}$$

Mathematically, let $E_n$ be an $B \times B$ matrix such that its $n$-th column and the $n$-th row comprises ones, while all other entries are zero. We add the matrix $\log \zeta \times E_n$ to the similarity matrix. Then $S_{*,n}$ becomes $S_{*,n} + \log \zeta$. The loss function based on the revised similarity matrix becomes:

$$\begin{aligned}
L_{\text{T2I},\lambda}^m((x^T, x^I), S; \theta) &= \sum_{\substack{k \in [B] \\ k \neq n}} -\log \frac{e^{S_{k,k}}}{\sum_{\substack{j \in [B] \\ j \neq n}} e^{S_{k,j}} + e^{\log \zeta} \cdot e^{S_{k,n}}} + \text{Pos}\left((x^T, x^I); \theta\right) \\
&= \sum_{\substack{k \in [B] \\ k \neq n}} -\log \frac{e^{S_{k,k}}}{\sum_{\substack{j \in [B] \\ j \neq n}} e^{S_{k,j}} + \zeta \cdot e^{S_{k,n}}} + \text{Pos}\left((x^T, x^I); \theta\right) \\
&= \sum_{\substack{k \in [B] \\ k \neq n}} -\log \frac{e^{S_{k,k}}}{\sum_{j \in [B]} e^{S_{k,j}} + (\zeta - 1) \cdot e^{S_{k,n}}} + \text{Pos}\left((x^T, x^I); \theta\right)
\end{aligned}$$

We can easily see that as $\zeta$ approaches 0, the loss function $L_{\text{T2I},\zeta}^m$ converges to $L_{\text{T2I, -neg}}^m$ in (9). When $\zeta = 1$, the loss function equals the original one. To separate the change in the $\zeta$ approaching 0 from 1 process, we perform a Taylor expansion at $\zeta = 0$ and drop the $O((\zeta - 1)^2)$ term, then $L_{\text{T2I},\zeta}^m$ becomes

$$\sum_{\substack{k \in [B] \\ k \neq n}} -\log \frac{e^{S_{k,k}}}{\sum_{j \in [B]} e^{S_{k,j}}} + (\zeta - 1) \cdot \sum_{\substack{k \in [B] \\ k \neq n}} \left( \frac{\sum_{j \in [B]} e^{S_{k,j}}}{e^{S_{k,n}}} \right) + O((\zeta - 1)^2) + \text{Pos}((x^T, x^I); \theta).$$

And by setting $\zeta = 0$, the loss function $L_{\text{T2I},0}^m$ indicates that the influence of $(x^T, x^I)$ when it serves as the negative sample is fully removed from the training process.

$$\begin{aligned}
L_{\text{T2I},0}^m &= \sum_{\substack{k \in [B] \\ k \neq n}} -\log \frac{e^{S_{k,k}}}{\sum_{j \in [B]} e^{S_{k,j}}} + (0 - 1) \cdot \sum_{\substack{k \in [B] \\ k \neq n}} \left( \frac{\sum_{j \in [B]} e^{S_{k,j}}}{e^{S_{k,n}}} \right) + \text{Pos}((x^T, x^I); \theta) \\
&= L_{\text{T2I}}^m(S; \theta) - \sum_{\substack{k \in [B] \\ k \neq n}} \left( \frac{\sum_{j \in [B]} e^{S_{k,j}}}{e^{S_{k,n}}} \right).
\end{aligned}$$

**Single Data Pair Version.** From above discussion, to quantify the impact of $x^T$ and $x^I$ as negative samples, we first define $\text{Neg}\left((x^T, x^I); \theta\right)$ as

$$\text{Neg}\left((x^T, x^I); \theta\right) = \sum_{\substack{k \in [B] \\ k \neq n}} \left( \frac{\sum_{j \in [B]} e^{S_{k,j}}}{e^{S_{k,n}}} + \frac{\sum_{j \in [B]} e^{S_{j,k}}}{e^{S_{n,k}}} \right),$$

Down-weighting the influence as a negative sample by $\zeta$ from 1 to 0, this influence in the loss function is then approximately eliminated. In this process, the loss function becomes

$$L_{\text{Total},\zeta}(\theta) = \sum_{(U,V) \in \mathcal{B}} L_{\text{Batch}}(U, V; \theta) + \frac{\delta}{2} \|\theta\|_2^2 + (\zeta - 1) \cdot \text{Neg}((x^T, x^I); \theta).$$

And the parameters are obtained by $\hat{\theta}_\zeta = \arg\min_\theta L_{\text{Total},\zeta}(\theta)$. From this minimizing condition, we have

$$\sum_{(U,V)\in\mathcal{B}} \nabla_\theta L_{\text{Batch}}(U,V;\hat{\theta}_\zeta) + (\zeta-1)\cdot\nabla_\theta\text{Neg}((x^T,x^I);\hat{\theta}_\zeta) = 0$$

Perform a Taylor expand at $\theta = \hat{\theta}$, we have

$$\sum_{(U,V)\in\mathcal{B}} \nabla_\theta L_{\text{Batch}}(U,V;\hat{\theta}) + (\zeta-1)\cdot\nabla_\theta\text{Neg}((x^T,x^I);\hat{\theta}) + \sum_{(U,V)\in\mathcal{B}} \nabla_\theta^2 L_{\text{Batch}}(U,V;\hat{\theta})\cdot\left(\hat{\theta}_\zeta - \hat{\theta}\right) \approx 0$$

Because $\hat{\theta}$ minimizes $\sum_{(U,V)\in\mathcal{B}} L_{\text{Batch}}(U,V;\hat{\theta})$, the first term in the above equation equals $0$. Then

$$\hat{\theta}_\zeta - \hat{\theta} = -(\zeta-1)\cdot H_{\hat{\theta}}^{-1}\cdot\nabla_\theta\text{Neg}((x^T,x^I);\hat{\theta})$$

where $H_{\hat{\theta}} = \nabla_\theta^2 \sum_{(U,V)\in\mathcal{B}} L_{\text{Batch}}(U,V;\hat{\theta}) + \delta I$ is the Hessian matrix at $\hat{\theta}$.

$$\text{negative-IF}((x^T,x^I);\hat{\theta}) \triangleq \left.\frac{d\hat{\theta}_\zeta}{d\zeta}\right|_{\zeta=0} = -H_{\hat{\theta}}^{-1}\cdot\nabla_\theta\text{Neg}((x^T,x^I);\hat{\theta})$$

**Extension to Multiple Samples.** Then, we extend the above influence evaluation to a subset $\mathcal{D}^* \subset \mathcal{D}$. Let set $S$ to index the batches containing data from $\mathcal{D}^*$. For every $m \in S$, define an index set $E_m$ to specify the position of data from $\mathcal{D}^*$ within the $m$-th batch. We encapsulate the assigned results as $\text{Seg} = \{(m, E_m)|m \in S\}$. By employing a derivation method similar to that used for a single data point, we can obtain the parameter-related influence function for $\mathcal{D}^*$.

**Proposition D.2.** *The influence function for dataset $\mathcal{D}^*$ serving as negative sample (negative-IF) is*

$$Neg(\mathcal{D}^*, Seg; \theta) = \sum_{m\in S}\sum_{k\in[B]/E_m}\left(\frac{\sum_{j\in[B]}e^{S_{k,j}}}{\sum_{n\in E_m}e^{S_{k,n}}} + \frac{\sum_{j\in[B]}e^{S_{j,k}}}{\sum_{n\in E_m}e^{S_{n,k}}}\right)$$

*And*

$$\text{negative-IF}(\mathcal{D}^*, Seg; \hat{\theta}) = -H_{\hat{\theta}}^{-1}\cdot\nabla_\theta Neg(\mathcal{D}^*, Seg; \hat{\theta}).$$

*Proof.* $\text{Seg} = \{(m, E_m)|m \in S\}$, for $m \in S$, $U_m, V_m$ are the text and image embedding for the $m$-th batch, respectively. For $n \in E_m$, $u_n$ and $v_n$ are embeddings for a single data pair in $m$-th batch, which is included in the dataset to be evaluated $\mathcal{D}^*$.

**Step 1**. Noting the data in $\mathcal{D}^*$ may come from different batches and multiple data from one batch, then we firstly derive the loss function approximation with separated negative sample influence removed.

For the $m$-th batch, $m \in S$, after removing the impact of the data indexed by $E_m$ as a negative sample, the loss function corresponding to this batch should become:

$$L_{\text{T2I, -neg}}^m((x^T,x^I),S;\theta) = \sum_{\substack{k\in[B]\\k\notin E_m}}\frac{e^{S_{k,k}}}{\sum_{\substack{j\in[B]\\j\notin E_m}}e^{S_{k,j}}} + \text{Pos}((x^T,x^I);\theta). \tag{10}$$

Then, for $n \in E_m$, let $E_n$ be an $B \times B$ matrix such that its $n$-th column and the $n$-th row comprises ones, while all other entries are zero. We add the matrix $\log\zeta \times E_n$ to the similarity matrix. Then, the loss function based on the revised similarity matrix becomes:

$$L_{\text{T2I},\lambda}^m((x^T,x^I),S;\theta) = \sum_{\substack{k\in[B]\\k\notin E_m}}-\log\frac{e^{S_{k,k}}}{\sum_{j\in[B]}e^{S_{k,j}} + (\zeta-1)\cdot\sum_{n\in E_m}e^{S_{k,n}}} + \text{Pos}((x^T,x^I);\theta).$$

We can easily see that as $\zeta$ approaches $0$, the loss function $L_{\text{T2I},\zeta}^m$ converges to $L_{\text{T2I, -neg}}^m$ in (9). When $\zeta = 1$, the loss function equals the original one. To separate the change in the $\zeta$ approaching $0$

from 1 process, we perform a Taylor expansion at $\zeta = 0$ and drop the $O((\zeta - 1)^2)$ term, then $L_{\text{T2I},\zeta}^m$ becomes

$$\sum_{\substack{k \in [B] \\ k \notin E_m}} -\log \frac{e^{S_{k,k}}}{\sum_{j \in [B]} e^{S_{k,j}}} + (\zeta - 1) \cdot \sum_{\substack{k \in [B] \\ k \notin E_m}} \left( \frac{\sum_{j \in [B]} e^{S_{k,j}}}{\sum_{n \in E_m} e^{S_{k,n}}} \right) + O((\zeta - 1)^2) + \text{Pos}((x^T, x^I); \theta).$$

And by setting $\zeta = 0$, the loss function $L_{\text{T2I},0}^m$ indicates that the influence of $(x^T, x^I)$ when it serves as the negative sample is fully removed from the training process.

$$L_{\text{T2I},0}^m = \sum_{\substack{k \in [B] \\ k \notin E_m}} -\log \frac{e^{S_{k,k}}}{\sum_{j \in [B]} e^{S_{k,j}}} + (0 - 1) \cdot \sum_{\substack{k \in [B] \\ k \notin E_m}} \left( \frac{\sum_{j \in [B]} e^{S_{k,j}}}{\sum_{n \in E_m} e^{S_{k,n}}} \right) + \text{Pos}((x^T, x^I); \theta)$$

$$= L_{\text{T2I}}^m(S; \theta) - \sum_{\substack{k \in [B] \\ k \notin E_m}} \left( \frac{\sum_{j \in [B]} e^{S_{k,j}}}{\sum_{n \in E_m} e^{S_{k,n}}} \right)$$

By down-weighting the influence of $\mathcal{D}^*$ as negative samples by $\zeta$, the total loss function becomes

$$L_{\text{Total},\zeta}(\theta) = \sum_{(U,V) \in \mathcal{B}} L_{\text{Batch}}(U, V; \theta) + \frac{\delta}{2} \|\theta\|_2^2 + (\zeta - 1) \cdot \sum_{m \in S} \sum_{k \in [B]/E_m} \left( \frac{\sum_{j \in [B]} e^{S_{k,j}}}{\sum_{n \in E_m} e^{S_{k,n}}} \right)$$

Then denote $\text{Neg}(\mathcal{D}^*, \text{Seg}; \theta)$ as

$$\text{Neg}(\mathcal{D}^*, \text{Seg}; \theta) = \sum_{m \in S} \sum_{k \in [B]/E_m} \left( \frac{\sum_{j \in [B]} e^{S_{k,j}}}{\sum_{n \in E_m} e^{S_{k,n}}} + \frac{\sum_{j \in [B]} e^{S_{j,k}}}{\sum_{n \in E_m} e^{S_{n,k}}} \right)$$

And the loss function with the negative-sample influence of $\mathcal{D}^*$ explicitly removed is

$$L_{\text{Total},0}(\theta) = \sum_{(U,V) \in \mathcal{B}} L_{\text{Batch}}(U, V; \theta) + \frac{\delta}{2} \|\theta\|_2^2 - \text{Neg}(\mathcal{D}^*, \text{Seg}; \theta)$$

**Step 2**. The parameters are obtained by $\hat{\theta}_\zeta = \arg\min_\theta L_{\text{Total},\zeta}(\theta)$. From this minimizing condition, we have

$$\sum_{(U,V) \in \mathcal{B}} \nabla_\theta L_{\text{Batch}}(U, V; \hat{\theta}_\zeta) + (\zeta - 1) \cdot \nabla_\theta \text{Neg}(\mathcal{D}^*, \text{Seg}; \hat{\theta}) = 0$$

Perform a Taylor expand at $\theta = \hat{\theta}$, we have

$$\sum_{(U,V) \in \mathcal{B}} \nabla_\theta L_{\text{Batch}}(U, V; \hat{\theta}) + (\zeta - 1) \cdot \nabla_\theta \text{Neg}(\mathcal{D}^*, \text{Seg}; \hat{\theta}) + \sum_{(U,V) \in \mathcal{B}} \nabla_\theta^2 L_{\text{Batch}}(U, V; \hat{\theta}) \cdot \left( \hat{\theta}_\zeta - \hat{\theta} \right) \approx 0$$

Because $\hat{\theta}$ minimizes $\sum_{(U,V) \in \mathcal{B}} L_{\text{Batch}}(U, V; \hat{\theta})$, the first term in the above equation equals 0.

$$\text{negative-IF}(\mathcal{D}^*, \text{Seg}; \hat{\theta}) \triangleq \left. \frac{d\hat{\theta}_\zeta}{d\zeta} \right|_{\zeta=0} = -H_{\hat{\theta}}^{-1} \cdot \nabla_\theta \text{Neg}(\mathcal{D}^*, \text{Seg}; \hat{\theta})$$

where $H_{\hat{\theta}} = \nabla_\theta^2 \sum_{(U,V) \in \mathcal{B}} L_{\text{Batch}}(U, V; \hat{\theta}) + \delta I$ is the Hessian matrix at $\hat{\theta}$. $\qquad\square$

## E   APPROXIMATION ERROR BOUND

In the previous discussion, we have established that when applying the influence function method to contrastive learning, it is impractical to design a sample-specific up-weighting scheme that approximates the corresponding loss function resulting from the removal of a single pair in the batch without affecting the remaining data. Therefore, based on the previous derivation, we provide an estimation function $L^-$ for this loss function. Consider the dataset $\mathcal{D}^*$, define

$$L'(\mathcal{D}^*, \text{Seg}; \theta) \triangleq \text{Pos}(\mathcal{D}^*, \text{Seg}; \theta) + \cdot \text{Neg}(\mathcal{D}^*, \text{Seg}; \theta),$$

Then the loss function with the influence of $\mathcal{D}^*$ removed becomes

$$L^-(\mathcal{B}, \mathcal{D}^*, \text{Seg}; \theta) = L_{Total}(\mathcal{B}; \theta) - L'(\mathcal{D}^*, \text{Seg}; \theta). \tag{11}$$

Equation equation 11 is based on the assumption that the influence of data acting as positive and negative samples on model parameters can be linearly superimposed, and we can leverage ECIF to edit the model based on the following corollary. This approach enables us to achieve the unlearning or updating of specific data without the need to remove data and retrain the model.

Assume $\hat{\theta} = \arg\min L_{Total}$ is the original model parameter, and $\hat{\theta}(-\mathcal{D}^*)$ is the minimizer of $L^-$, which is obtained from retraining. Denote $\theta_{if}(-\mathcal{D}^*)$ as the updated model with the influence of $\mathcal{D}^*$ removed and is obtained by the ECIF method, which is an estimation for $\hat{\theta}(-\mathcal{D}^*)$. Because we concentrate on $\mathcal{D}^*$, we omit the Seg in the above definitions for short.

In this part, we will study the error between the estimated influence given by the ECIF method and retraining. We use the parameter changes as the evaluation metric:

$$\left| \left( \theta_{if}(-\mathcal{D}^*) - \hat{\theta} \right) - \left( \hat{\theta}(-\mathcal{D}^*) - \hat{\theta} \right) \right| = \left| \theta_{if}(-\mathcal{D}^*) - \hat{\theta}(-\mathcal{D}^*) \right| \tag{12}$$

Before our main theorem of the upper bound for equation (12), we need to prove corollaries and make some assumptions.

**Proposition E.1.** *Assume that influence as positive sample and as negative sample can be linearly superposed. Then when the influence of dataset $\mathcal{D}^*$ as positive sample is up-weighted by $\epsilon$ and that as negative sample is up-weighted by $\zeta$, then the loss function become*

$$L^-(\mathcal{D}^*, Seg; \theta; \epsilon, \zeta) \triangleq L_{Total}(\mathcal{B}; \theta) + \epsilon \cdot \nabla_\theta Pos(\mathcal{D}^*, Seg; \hat{\theta}) + (\zeta - 1) \cdot Neg(\mathcal{D}^*, Seg; \hat{\theta})$$

*And corresponding parameters $\theta_{\epsilon, \zeta}$ are defined as*

$$\hat{\theta}_{\epsilon, \zeta}(-\mathcal{D}^*) = \arg\min_\theta L^-(\mathcal{D}^*, Seg; \theta; \epsilon, \zeta)$$

*The approximation of $\hat{\theta}_{\epsilon, \zeta}(-\mathcal{D}^*)$ is derived as*

$$\hat{\theta}_{\epsilon, \zeta}(\mathcal{D}^*) \approx \theta_{\epsilon, \zeta}(\mathcal{D}^*) = \hat{\theta} - H_{\hat{\theta}}^{-1} \cdot \left( \frac{\sqrt{2}}{2} \cdot \nabla_\theta Pos(\mathcal{D}^*, Seg; \hat{\theta}) + \frac{\sqrt{2}}{2} \cdot \nabla_\theta Neg(\mathcal{D}^*, Seg; \hat{\theta}) \right) \tag{13}$$

**Property E.2.** *Assume that influence as positive sample and as negative sample can be linearly superposed. Then when the influence of dataset $\mathcal{D}^*$ as positive sample is up-weighted by $\epsilon$ and that as negative sample is up-weighted by $\zeta$, then the loss function become*

$$L^-(\mathcal{D}^*, Seg; \theta; \epsilon, \zeta) \triangleq L_{Total}(\mathcal{B}; \theta) + \epsilon \cdot \nabla_\theta Pos(\mathcal{D}^*, Seg; \hat{\theta}) + (\zeta - 1) \cdot Neg(\mathcal{D}^*, Seg; \hat{\theta})$$

*And corresponding parameters $\theta_{\epsilon, \zeta}$ are defined as*

$$\hat{\theta}_{\epsilon, \zeta}(-\mathcal{D}^*) = \arg\min_\theta L^-(\mathcal{D}^*, Seg; \theta; \epsilon, \zeta)$$

*The approximation of $\hat{\theta}_{\epsilon, \zeta}(-\mathcal{D}^*)$ is derived as*

$$\begin{aligned}
\hat{\theta}_{\epsilon, \zeta}(-\mathcal{D}^*) \approx & \theta_{\epsilon, \zeta}(-\mathcal{D}^*) \\
\triangleq & \hat{\theta} - H_{\hat{\theta}}^{-1} \cdot \left( \epsilon \cdot \nabla_\theta Pos(\mathcal{D}^*, Seg; \hat{\theta}) + (\zeta - 1) \cdot \nabla_\theta Neg(\mathcal{D}^*, Seg; \hat{\theta}) \right)
\end{aligned} \tag{14}$$

*Proof.* Assume that influence as positive sample and as negative sample can be linearly superposed. Then when the influence of dataset $\mathcal{D}^*$ as positive sample is up-weighted by $\epsilon$ and that as negative sample is up-weighted by $\zeta$, then the loss function become

$$L^-(\mathcal{D}^*, \text{Seg}; \theta; \epsilon, \zeta) \triangleq L_{Total}(\mathcal{B}; \theta) + \epsilon \cdot \text{Pos}(\mathcal{D}^*, \text{Seg}; \hat{\theta}) + (\zeta - 1) \cdot \text{Neg}(\mathcal{D}^*, \text{Seg}; \hat{\theta})$$

And corresponding parameters $\theta_{\epsilon, \zeta}$ are defined as

$$\hat{\theta}_{\epsilon, \zeta}(-\mathcal{D}^*) = \arg\min_\theta L^-(\mathcal{D}^*, \text{Seg}; \theta; \epsilon, \zeta)$$

Then, from the minimizing condition,

$$\nabla_\theta L_{Total}(\mathcal{B}; \hat{\theta}_{\epsilon,\zeta}) + \epsilon \cdot \nabla_\theta \text{Pos}(\mathcal{D}^*, \text{Seg}; \hat{\theta}_{\epsilon,\zeta}) + (\zeta - 1) \cdot \nabla_\theta \text{Neg}(\mathcal{D}^*, \text{Seg}; \hat{\theta}_{\epsilon,\zeta}) = 0,$$

where $\hat{\theta}_{\epsilon,\zeta}(-\mathcal{D}^*)$ is written as $\hat{\theta}_{\epsilon,\zeta}$ for short. Perform a Taylor expansion around $\theta = \hat{\theta}$, then we have

$$\nabla_\theta L_{Total}(\mathcal{B}; \hat{\theta}) + \epsilon \cdot \nabla_\theta \text{Pos}(\mathcal{D}^*, \text{Seg}; \hat{\theta}) + (\zeta - 1) \cdot \nabla_\theta \text{Neg}(\mathcal{D}^*, \text{Seg}; \hat{\theta})$$
$$+ \nabla_\theta^2 L_{Total}(\mathcal{B}; \hat{\theta}) \cdot \left( \hat{\theta}_{\epsilon,\zeta} - \hat{\theta} \right) = 0.$$

Because $\hat{\theta}$ minimizes $L_{Total}(\mathcal{B}; \theta)$, the first term in above equation equals 0. Then we have

$$\hat{\theta}_{\epsilon,\zeta} \approx \hat{\theta} - H_{\hat{\theta}}^{-1} \cdot \left( \epsilon \cdot \nabla_\theta \text{Pos}(\mathcal{D}^*, \text{Seg}; \hat{\theta}) + (\zeta - 1) \cdot \nabla_\theta \text{Neg}(\mathcal{D}^*, \text{Seg}; \hat{\theta}) \right)$$
$$= \hat{\theta} - \epsilon \cdot H_{\hat{\theta}}^{-1} \cdot \nabla_\theta \text{Pos}(\mathcal{D}^*, \text{Seg}; \hat{\theta}) - (\zeta - 1) \cdot H_{\hat{\theta}}^{-1} \cdot \nabla_\theta \text{Neg}(\mathcal{D}^*, \text{Seg}; \hat{\theta})$$
$$= \hat{\theta} + \epsilon \cdot \text{positive-IF}(\mathcal{D}^*, \text{Seg}; \hat{\theta}) + (\zeta - 1) \cdot \text{negative-IF}(\mathcal{D}^*, \text{Seg}; \hat{\theta})$$

where $H_{\hat{\theta}} = \nabla_\theta^2 \sum_{(U,V) \in \mathcal{B}} L_{\text{Batch}}(U, V; \hat{\theta}) + \delta I$. When $\epsilon = -1, \zeta = 0$, $\hat{\theta}_{-1,0}$ estimates the parameters obtained by retraining after $\mathcal{D}^*$ removed.

$\square$

**Assumption E.3.** The loss $L_{\text{Batch}}(U, V, \theta)$ is convex and twice-differentiable in $\theta$, with positive regularization $\delta > 0$. There exists $C_H \in \mathbb{R}$ such that

$$\|\nabla_\theta^2 L_{\text{Batch}}(U, V; \theta_1) - \nabla_\theta^2 L_{\text{Batch}}(U, V; \theta_2)\|_2 \leq C_H \|\theta_1 - \theta_2\|_2$$

for all $(U, V) \in \mathcal{B}$ and $\theta_1, \theta_2 \in \Theta$.

**Assumption E.4.** The function $L'((x^T, x^I); \theta)$:

$$L'((x^T, x^I); \theta) = \text{Pos}((x^T, x^I); \theta) + \text{Neg}((x^T, x^I); \theta)$$

is convex and twice-differentiable in $\theta$, with some positive regularization. There exists $C_H' \in \mathbb{R}$ such that

$$\|\nabla_\theta^2 L'((x^T, x^I); \theta_1) - \nabla_\theta^2 L'((x^T, x^I); \theta_2)\|_2 \leq C_H' \|\theta_1 - \theta_2\|_2$$

for all $(x^T, x^I) \in \mathcal{D}^*$ and $\theta_1, \theta_2 \in \Theta$.

**Corollary E.5.**

$$\|\nabla_\theta^2 L^-(\mathcal{D}^*, Seg; \theta_1) - \nabla_\theta^2 L^-(\mathcal{D}^*, Seg; \theta_2)\|_2 \leq (|\mathcal{B}| \cdot C_H + |\mathcal{D}^*| \cdot C_H'|) \|\theta_1 - \theta_2\|$$

*Define* $C_H^- \triangleq |\mathcal{B}| \cdot C_H + |\mathcal{D}^*| \cdot C_H'$

**Definition E.6.** Define $|\mathcal{D}|$ as the number of pairs

$$C_L' = \max_{(x^T, x^I) \in \mathcal{B}} \left\| \nabla_\theta L'((x^T, x^I); \hat{\theta}) \right\|_2,$$

$$\sigma'_{\min} = \text{smallest singular value of } \nabla_\theta^2 L^-(\mathcal{D}^*, \text{Seg}; \hat{\theta}),$$

$$\sigma_{\min} = \text{smallest singular value of } \nabla_\theta^2 L_{\text{Total}}(\mathcal{B}; \hat{\theta}),$$

Based on above corollaries and assumptions, we derive the following theorem.

**Theorem E.7.** *We obtain the error between the actual influence and our predicted influence as follows:*

$$\left\| \hat{\theta}(-\mathcal{D}^*) - \theta_{if}(-\mathcal{D}^*) \right\|$$
$$\leq \frac{C_H' C_H^- |\mathcal{D}^*|^2 {C_L'}^2}{2(\sigma'_{\min} + \delta)^3} + \left| \frac{2\delta + \sigma_{\min} + \sigma'_{\min}}{(\delta + \sigma'_{\min}) \cdot (\delta + \sigma_{\min})} \right| \cdot C_L' |\mathcal{D}^*|$$

*Proof.* We will use the one-step Newton approximation as an intermediate step. Define $\Delta\theta_{Nt}(-\mathcal{D}^*)$ as

$$\Delta\theta_{Nt}(-\mathcal{D}^*) \triangleq H_\delta^{-1} \cdot \nabla_\theta L'(\mathcal{D}^*, \text{Seg}; \hat\theta),$$

where $H_\delta = \delta \cdot I + \nabla_\theta^2 L^-(\mathcal{D}^*, \text{Seg}; \hat\theta)$ is the regularized empirical Hessian at $\hat\theta$ but reweighed after removing the influence of $\mathcal{D}^*$. Then the one-step Newton approximation for $\hat\theta(-\mathcal{D}^*)$ is defined as $\theta_{Nt}(-\mathcal{D}^*) \triangleq \Delta\theta_{Nt}(-\mathcal{D}^*) + \hat\theta$.

In the following, we will separate the error between $\theta_{if}(-\mathcal{D}^*)$ and $\hat\theta(-\mathcal{D}^*)$ into the following two parts:

$$\hat\theta(-\mathcal{D}^*) - \theta_{if}(-\mathcal{D}^*) = \underbrace{\hat\theta(-\mathcal{D}^*) - \theta_{Nt}(-\mathcal{D}^*)}_{\text{Err}_{\text{Nt, act}}(-\mathcal{D}^*)} + \underbrace{\left(\theta_{Nt}(-\mathcal{D}^*) - \hat\theta\right) - \left(\theta_{if}(-\mathcal{D}^*) - \hat\theta\right)}_{\text{Err}_{\text{Nt, if}}(-\mathcal{D}^*)}$$

Firstly, in **Step** 1, we will derive the bound for Newton-actual error $\text{Err}_{\text{Nt, act}}(-\mathcal{D}^*)$. Since $L^-(\theta)$ is strongly convex with parameter $\sigma'_{\min} + \delta$ and minimized by $\hat\theta(-\mathcal{D}^*)$, we can bound the distance $\left\|\hat\theta(-\mathcal{D}^*) - \theta_{Nt}(-\mathcal{D}^*)\right\|_2$ in terms of the norm of the gradient at $\theta_{Nt}$:

$$\left\|\hat\theta(-\mathcal{D}^*) - \theta_{Nt}(-\mathcal{D}^*)\right\|_2 \le \frac{2}{\sigma'_{\min} + \delta} \left\|\nabla_\theta L^-\left(\theta_{Nt}(-\mathcal{D}^*)\right)\right\|_2 \tag{15}$$

Therefore, the problem reduces to bounding $\left\|\nabla_\theta L^-\left(\theta_{Nt}(-\mathcal{D}^*)\right)\right\|_2$. Noting that $\nabla_\theta L'(\hat\theta) = -\nabla_\theta L^-$. This is because $\hat\theta$ minimizes $L^- + L'$, that is,

$$\nabla_\theta L^-(\hat\theta) + \nabla_\theta L'(\hat\theta) = 0.$$

Recall that $\Delta\theta_{Nt} = H_\delta^{-1} \cdot \nabla_\theta L'(\mathcal{D}^*, \text{Seg}; \hat\theta) = -H_\delta^{-1} \cdot \nabla_\theta L^-(\mathcal{D}^*, \text{Seg}; \hat\theta)$. Given the above conditions, we can have this bound for $\text{Err}_{\text{Nt, act}}(-\mathcal{D}^*)$.

$$
\begin{aligned}
&\left\|\nabla_\theta L^-\left(\theta_{Nt}(-\mathcal{D}^*)\right)\right\|_2 \\
&= \left\|\nabla_\theta L^-\left(\hat\theta + \Delta\theta_{Nt}(-\mathcal{D}^*)\right)\right\|_2 \\
&= \left\|\nabla_\theta L^-\left(\hat\theta + \Delta\theta_{Nt}(-\mathcal{D}^*)\right) - \nabla_\theta L^-\left(\hat\theta\right) - \nabla_\theta^2 L^-\left(\hat\theta\right) \cdot \Delta\theta_{Nt}(-\mathcal{D}^*)\right\|_2 \\
&= \left\|\int_0^1 \left(\nabla_\theta^2 L^-\left(\hat\theta + t\cdot\Delta\theta_{Nt}(-\mathcal{D}^*)\right) - \nabla_\theta^2 L^-\left(\hat\theta\right)\right)\Delta\theta_{Nt}(-\mathcal{D}^*)\, dt\right\|_2 \\
&\le \frac{C_H^-}{2}\|\Delta\theta_{Nt}(-\mathcal{D}^*)\|_2^2 = \frac{C_H^-}{2}\left\|\left[\nabla_\theta^2 L^-(\hat\theta)\right]^{-1}\nabla_\theta L^-(\hat\theta)\right\|_2^2 \\
&\le \frac{C_H^-}{2(\sigma'_{\min}+\delta)^2}\left\|\nabla_\theta L^-(\hat\theta)\right\|_2^2 = \frac{C_H^-}{2(\sigma'_{\min}+\delta)^2}\left\|\nabla_\theta L'(\hat\theta)\right\|_2^2 \\
&\le \frac{C_H^-\|\mathcal{D}^*\|^2 C_L'^2}{2(\sigma'_{\min}+\delta)^2}.
\end{aligned}
\tag{16}
$$

Now we come to **Step** 2 to bound $\text{Err}_{\text{Nt, if}}(-\mathcal{D}^*)$, and we will bound the difference in parameter change between Newton and our ECIF method.

$$
\begin{aligned}
&\left\|\left(\theta_{Nt}(-\mathcal{D}^*) - \hat\theta\right) - \left(\theta_{if}(-\mathcal{D}^*) - \hat\theta\right)\right\| \\
&= \left\|\left[\left(\delta\cdot I + \nabla_\theta^2 L^-\left(\hat\theta\right)\right)^{-1} + \left(\delta\cdot I + \nabla_\theta^2 L_{\text{Total}}\left(\hat\theta\right)\right)^{-1}\right]\cdot\nabla_\theta L'(\mathcal{D}^*, \text{Seg}; \hat\theta)\right\|
\end{aligned}
$$

For simplification, we use matrix $A$, $B$ for the following substitutions:

$$A = \delta\cdot I + \nabla_\theta^2 L^-\left(\hat\theta\right)$$

$$B = \delta\cdot I + \nabla_\theta^2 L_{\text{Total}}\left(\hat\theta\right)$$

And $A$ and $B$ are positive definite matrices with the following properties

$$\delta + \sigma'_{\min} \prec A \prec \delta + \sigma'_{\max}$$
$$\delta + \sigma_{\min} \prec B \prec \delta + \sigma_{\max}$$

Therefore, we have

$$
\begin{aligned}
&\left\| \left( \theta_{Nt}(-\mathcal{D}^*) - \hat{\theta} \right) - \left( \theta_{if}(-\mathcal{D}^*) - \hat{\theta} \right) \right\| \\
&= \left\| \left( A^{-1} + B^{-1} \right) \cdot \nabla_\theta L^-(\mathcal{D}^*, \text{Seg}; \hat{\theta}) \right\| \\
&\leq \left\| A^{-1} + B^{-1} \right\| \cdot \left\| \nabla_\theta L^-(\mathcal{D}^*, \text{Seg}; \hat{\theta}) \right\| \\
&\leq \left| \frac{2\delta + \sigma_{\min} + \sigma'_{\min}}{(\delta + \sigma'_{\min}) \cdot (\delta + \sigma_{\min})} \right| \cdot \left\| \nabla_\theta L^-(\mathcal{D}^*, \text{Seg}; \hat{\theta}) \right\| \\
&\leq \left| \frac{2\delta + \sigma_{\min} + \sigma'_{\min}}{(\delta + \sigma'_{\min}) \cdot (\delta + \sigma_{\min})} \right| \cdot C'_L |\mathcal{D}^*|
\end{aligned}
\tag{17}
$$

By combining the conclusions from Step I and Step II in Equations 15, 16 and 17, we obtain the error between the actual influence and our predicted influence as follows:

$$
\begin{aligned}
&\left\| \hat{\theta}(-\mathcal{D}^*) - \theta_{if}(-\mathcal{D}^*) \right\| \\
&\leq \frac{C'_H C_H^- |\mathcal{D}^*|^2 C'^2_L}{2(\sigma'_{\min} + \delta)^3} + \left| \frac{2\delta + \sigma_{\min} + \sigma'_{\min}}{(\delta + \sigma'_{\min}) \cdot (\delta + \sigma_{\min})} \right| \cdot C'_L |\mathcal{D}^*|.
\end{aligned}
$$

It is notable that such error bound is small when the number of removal samples $|\mathcal{D}^*|$ is fixed as in practice $\delta = O(|\mathcal{B}|)$. $\qquad\square$

## F  APPLICATIONS OF ECIF

### F.1  TASK-RELATED IS

**Property F.1.** *Considering a specific set $\mathcal{D}'$ with text and image embeddings $U'$ and $V'$, and a dataset $D^*$ to be removed, then we have*

$$
\begin{aligned}
&L_{Batch}(U', V'; \hat{\theta}(-D^*)) - L_{Batch}(U', V'; \hat{\theta}) \approx \nabla L_{Batch}(U', V'; \hat{\theta})^{\mathrm{T}} (\hat{\theta}(-D^*) - \hat{\theta}) \\
&= -\nabla L_{Batch}(U', V'; \hat{\theta})^{\mathrm{T}} \cdot \left( positive\text{-}IF(\mathcal{D}^*, Seg; \hat{\theta}) + negative\text{-}IF(\mathcal{D}^*, Seg; \hat{\theta}) \right).
\end{aligned}
\tag{18}
$$

*where $\hat{\theta}(-D^*)$ is the optimal model for the loss eliminating $D^*$, positive-IF$(\mathcal{D}^*; Seg; \hat{\theta})$ and negative-IF$(\mathcal{D}^*, Seg; \hat{\theta})$ are obtained from Proposition 4.3 for $D^*$.*

*Proof.*

$$
\begin{aligned}
\text{IS}(\mathcal{D}', \mathcal{D}^*; \text{Seg}) &\triangleq - \left. \frac{dL_{\text{Batch}}(U', V'; \theta_{\epsilon, \zeta=0})}{d\epsilon} \right|_{\epsilon=0} - \left. \frac{dL_{\text{Batch}}(U', V'; \theta_{\epsilon=0, \zeta})}{d\zeta} \right|_{\zeta=0} \\
&\approx - \nabla L_{\text{Batch}}(U', V'; \hat{\theta})^{\mathrm{T}} \cdot \left( \text{positive-IF}(\mathcal{D}^*, \text{Seg}; \hat{\theta}) + \text{negative-IF}(\mathcal{D}^*, \text{Seg}; \hat{\theta}) \right)
\end{aligned}
$$

$\qquad\square$

### F.2  RELATIVE INFLUENCE SCORE

**Proposition F.2.** *Define $I = [positive\text{-}IF(x; \hat{\theta}), negative\text{-}IF(x; \hat{\theta})]$. If the $2 \times 2$ matrix $I^{\mathrm{T}} \cdot I$ is irreversible, then the optimization problem*

$$
\arg \max_{x \in \mathcal{D}} \max_{\epsilon, \zeta} \left| L_{Batch}(U', V'; \hat{\theta} + \Delta\hat{\theta}_{\epsilon, \zeta}(x)) - L_{Batch}(U', V'; \hat{\theta}) \right| \quad s.t. \left\| \Delta\hat{\theta}_{\epsilon, \zeta}(x) \right\|^2 \leq \rho^2
\tag{19}
$$

*is equivalent to*

$$\arg\max_{x\in\mathcal{D}} \|\textit{negative-IF}(x;\hat{\theta})\|_2^{-1} \left| \nabla L_{\textit{Batch}}(U',V';\hat{\theta})^{\mathrm{T}} \cdot \textit{negative-IF}(x;\hat{\theta}) \right|.$$

*Else, $I^{\mathrm{T}} \cdot I$ is reversible, then the initial problem is equivalent to*

$$\arg\max_{x\in\mathcal{D}} \|\nabla L_{\textit{Batch}}(U',V';\hat{\theta})\|_2^{-1} \left| \nabla L_{\textit{Batch}}(U',V';\hat{\theta})^{\mathrm{T}} \cdot I \cdot \left[I^{\mathrm{T}} \cdot I\right]^{-1} \cdot I^{\mathrm{T}} \cdot \nabla L_{\textit{Batch}}(U',V';\hat{\theta}) \right|.$$

*Proof.* From (14), we have

$$\left| L_{\text{Batch}}(U',V';\hat{\theta} + \Delta\hat{\theta}_{\epsilon,\zeta}(x)) - L_{\text{Batch}}(U',V';\hat{\theta}) \right| \tag{20}$$

$$\approx \left| \nabla L((U',V');\hat{\theta})^{\mathrm{T}} \cdot \Delta\hat{\theta}_{\epsilon,\zeta}(x)) \right| \tag{21}$$

$$\approx \left| \nabla L((U',V');\hat{\theta})^{\mathrm{T}} \cdot \left( \epsilon \cdot \text{positive-IF}((x^T, x^I);\hat{\theta}) + (\zeta - 1)\,\text{negative-IF}((x^T, x^I);\hat{\theta}) \right) \right| \tag{22}$$

And still from (14), the constraint in parameter changes can be written as

$$\left\| \Delta\hat{\theta}_{\epsilon,\zeta}(x) \right\| \tag{23}$$

$$= \|\epsilon \cdot \text{positive-IF}((x^T, x^I);\hat{\theta}) + (\zeta - 1)\,\text{negative-IF}((x^T, x^I);\hat{\theta})\| \leq \rho \tag{24}$$

We can regard (22) as the inner product between vector $u \triangleq \nabla L((U',V');\hat{\theta})$ and vector $v \triangleq \epsilon \cdot \text{positive-IF} + (\zeta - 1)\,\text{negative-IF}$.

If positive-IF is not parallel to negative-IF, then the constraint in equation (23) becomes that vector $v$ is chosen from a ball of radius $\rho$. Otherwise, the constraint is equivalent to a constraint on the norm of a vector that is parallel to positive-IF or negative-IF. Therefore, we will proceed with a classification discussion based on whether positive-IF and negative-IF are parallel.

Firstly, we consider the $\parallel$ case. As is well known, the inner product of vectors reaches its extreme when the two vectors are parallel. We can choose $\epsilon$ and $\zeta$ freely to make vectors $v \parallel u$. Assume that there exists $c \in \mathbb{R}$ s.t.

$$[\text{positive-IF}, \text{negative-IF}] \cdot \begin{bmatrix} \epsilon \\ \zeta - 1 \end{bmatrix} = c \cdot \nabla L((U',V');\hat{\theta})$$

Denote $[\text{positive-IF}, \text{negative-IF}]$ as $I$

$$[\text{positive-IF}, \text{negative-IF}] \cdot \begin{bmatrix} \epsilon \\ \zeta - 1 \end{bmatrix} = c \cdot \nabla L((U',V');\hat{\theta})$$

$$\begin{bmatrix} \text{positive-IF}^{\mathrm{T}} \\ \text{negative-IF}^{\mathrm{T}} \end{bmatrix} \cdot [\text{positive-IF}, \text{negative-IF}] \cdot \begin{bmatrix} \epsilon \\ \zeta - 1 \end{bmatrix} = c \cdot \begin{bmatrix} \text{positive-IF}^{\mathrm{T}} \\ \text{negative-IF}^{\mathrm{T}} \end{bmatrix} \cdot \nabla L((U',V');\hat{\theta})$$

$$I^{\mathrm{T}} \cdot I \cdot \begin{bmatrix} \epsilon \\ \zeta - 1 \end{bmatrix} = c \cdot I^{\mathrm{T}} \cdot \nabla L((U',V');\hat{\theta})$$

$$\begin{bmatrix} \epsilon \\ \zeta - 1 \end{bmatrix} = c \cdot \left[I^{\mathrm{T}} \cdot I\right]^{-1} \cdot I^{\mathrm{T}} \cdot \nabla L((U',V');\hat{\theta})$$

Noting that $I^{\mathrm{T}} \cdot I$ is invertible matrix as long as positive-IF, negative-IF are not parallel. Considering the constraints of the length of vector $v$, then

$$\|c \cdot \nabla L((U',V');\hat{\theta})\| \leq \rho$$

We can make vector $v$ reach its largest norm with setting $c$ to an appropriate number:

$$c = \frac{\rho}{\|\nabla L((U',V');\hat{\theta})\|}$$

Finally, we obtain the expression of vector 2 that maximizes expression (20)

$$[\text{positive-IF}, \text{negative-IF}] \cdot \begin{bmatrix} \epsilon \\ \zeta - 1 \end{bmatrix} = c \cdot I \cdot \left[I^{\mathrm{T}} \cdot I\right]^{-1} \cdot I^{\mathrm{T}} \cdot \nabla L((U',V');\hat{\theta})$$

Then we have

$$
\left| L((U', V'); \theta_{\epsilon,\zeta}(x^T, x^I)) - L((U', V'); \hat{\theta}) \right|
$$

$$
= \left| \nabla L((U', V'); \hat{\theta})^{\mathrm{T}} \cdot \left( [\text{positive-IF}, \text{negative-IF}] \cdot \begin{bmatrix} \epsilon \\ \zeta - 1 \end{bmatrix} \right) \right|
$$

$$
= c \cdot \left| \nabla L((U', V'); \hat{\theta})^{\mathrm{T}} \cdot I \cdot \left[ I^{\mathrm{T}} \cdot I \right]^{-1} \cdot I^{\mathrm{T}} \cdot \nabla L((U', V'); \hat{\theta}) \right|
$$

where $I = [\text{positive-IF}, \text{negative-IF}]$.

$$
\arg \max_{(x^T, x^I) \in \mathcal{D}} \frac{\rho}{\|\nabla L((U', V'); \hat{\theta})\|} \cdot \left| \nabla L((U', V'); \hat{\theta})^{\mathrm{T}} \cdot I \cdot \left[ I^{\mathrm{T}} \cdot I \right]^{-1} \cdot I^{\mathrm{T}} \cdot \nabla L((U', V'); \hat{\theta}) \right|
$$

where $I = \left[ \text{positive-IF}((x^T, x^I); \hat{\theta}), \text{negative-IF}((x^T, x^I); \hat{\theta}) \right]$.

If positive-IF, negative-IF are not parallel, the optimization problem in form (19) is equivalent to

$$
\arg \max_{x \in \mathcal{D}} \frac{\rho}{\|\nabla L((U', V'); \hat{\theta})\|} \left| \nabla L((U', V'); \hat{\theta})^{\mathrm{T}} I \left[ I^{\mathrm{T}} \cdot I \right]^{-1} I^{\mathrm{T}} \nabla L((U', V'); \hat{\theta}) \right|.
$$

Because $\rho$ is independent of data, we can drop it and write the above equation as

$$
\arg \max_{x \in \mathcal{D}} \|\nabla L((U', V'); \hat{\theta})\|^{-1} \left| \nabla L((U', V'); \hat{\theta})^{\mathrm{T}} I \left[ I^{\mathrm{T}} \cdot I \right]^{-1} I^{\mathrm{T}} \nabla L((U', V'); \hat{\theta}) \right|.
$$

Then, we come to the second case where positive-IF $\parallel$ negative-IF. We can define a

$$
\left\| \Delta \hat{\theta}_{\epsilon,\zeta}(x) \right\| \tag{25}
$$

$$
= \| \epsilon \cdot \text{positive-IF}((x^T, x^I); \hat{\theta}) + (\zeta - 1) \text{negative-IF}((x^T, x^I); \hat{\theta}) \| \tag{26}
$$

$$
\triangleq \| \alpha(\epsilon, \zeta) \cdot \text{positive-IF}((x^T, x^I); \hat{\theta}) \| \leq \rho \tag{27}
$$

And the constraint is imposed on $\alpha$ by

$$
\alpha(\epsilon, \zeta) \leq \frac{\rho}{\|\text{positive-IF}((x^T, x^I); \hat{\theta})\|}
$$

Therefore, equation (20) is equivalent to

$$
\max_{\epsilon, \zeta} \left| \nabla L((U', V'); \hat{\theta})^{\mathrm{T}} \cdot \left( \alpha(\epsilon, \zeta) \cdot \text{positive-IF}((x^T, x^I); \hat{\theta}) \right) \right|
$$

$$
= \max_{\epsilon, \zeta} \alpha(\epsilon, \zeta) \cdot \left| \nabla L((U', V'); \hat{\theta})^{\mathrm{T}} \cdot \left( \text{positive-IF}((x^T, x^I); \hat{\theta}) \right) \right|
$$

$$
= \frac{\rho}{\|\text{positive-IF}((x^T, x^I); \hat{\theta}))\|} \cdot \left| \nabla L((U', V'); \hat{\theta})^{\mathrm{T}} \cdot \text{positive-IF}((x^T, x^I); \hat{\theta}) \right|
$$

$$
= \frac{\rho}{\|\text{negative-IF}((x^T, x^I); \hat{\theta})\|} \cdot \left| \nabla L((U', V'); \hat{\theta})^{\mathrm{T}} \cdot \text{negative-IF}((x^T, x^I); \hat{\theta}) \right|
$$

Because $\rho$ is independent of data, we can drop it and write the above equation as

$$
\|\text{positive-IF}((x^T, x^I); \hat{\theta}))\|^{-1} \cdot \left| \nabla L((U', V'); \hat{\theta})^{\mathrm{T}} \cdot \text{positive-IF}((x^T, x^I); \hat{\theta}) \right|
$$

$$
= \|\text{negative-IF}((x^T, x^I); \hat{\theta})\|^{-1} \cdot \left| \nabla L((U', V'); \hat{\theta})^{\mathrm{T}} \cdot \text{negative-IF}((x^T, x^I); \hat{\theta}) \right|.
$$

$\square$

# G  ADDITIONAL EXPERIMENTAL RESULTS

### G.1 DETAILS OF EXPERIMENT SETTINGS

**Datasets.** We employ four datasets for our utility and efficiency evaluation tasks, *FGVC-Aircraft dataset* (Maji et al., 2013), *Food101 dataset* (Bossard et al., 2014), *Flowers102 dataset* (Nilsback & Zisserman, 2008), and *Cifar-100 dataset* (Krizhevsky, 2009). For the misprediction traceback experiments, we focus on the first three datasets. The FGVC-Aircraft dataset comprises 10,000 images of airplanes, each annotated with the model and bounding box of the dominant aircraft depicted. The Food-101 dataset, publicly available for food image recognition, includes 101 food categories, with each category containing 1,000 images. The images feature food photographs captured from various angles and under different lighting conditions. The Flowers-102 dataset consists of 102 classes of flowers native to the United Kingdom, with each class containing between 40 and 258 images. We use *Cifar-10 dataset* (Krizhevsky et al., 2009) for the misalignment detection tasks.

**Implementation Details.** Our experiments utilized a Nvidia V100-32G GPU and 10 CPU cores with 64 GB memory. For all the following tasks, we employ the CLIP model 'ViT-B/16' and use LoRA few-shot learning.

For data influence removal, when our method is tested on a random sample removal task, 10% samples are randomly removed. For valuable (harmful) samples, we remove 10% of the valuable (harmful) data identified by ECIF. Each removal is repeated 3 times with different seeds.

For the experiment of *Identifying influential data for fine-tuning*, we first calculate the task-related IS for every individual sample and collect valuable data with positive IS, then choose to remove 30% of these. We conducted the experiments 3 times for each removal with different seeds. The experiment setting for *harmful data removal* is similar. In contrast, we select harmful data with negative IS. The experiments are carried out on the Food101, Flowers102, FGVC-Aircraft, and DTD datasets, and the remove ratio ranges from 0 to 90%. For each removal, we conducted the experiments 3 times with different seeds.

The *multiple samples removal* experiments are conducted on Food101, Flowers102, FGVC-Aircraft, and DTD datasets, with removal ratios from 1% to 7%, respectively.

For the *misprediction trace back* task, we conduct experiments on the Food101, Flowers102, FGVC-Aircraft, and DTD datasets. We first choose a mispredicted test sample as the target in algorithm 3, then calculate the relative IS for each individual sample in the training dataset. Noting that the relative IS is always positive. We visualize training samples with top-10 relative IS.

For the *misalignment detection* tasks, Cifar-10 and imagenette (smaller version of imageNet) datasets are used. We also applied standard data augmentation techniques on the training set, i.e., random cropping and random flipping. The model is optimized with Adam with weight decay (5e-1), and $\beta$ is set to 0.9. A dropout ratio of 0.25 is used. The training iterations are set to 30, with a learning rate of 2e-4 and a batch size of 16. The rank of the low-rank matrices of LoRA is set to 2. We trained the model on a poisoned version of the dataset (20% / 30% of the data samples are mislabeled). Then, we compute the influence score IS of all the training samples on the mispredicted test samples. In the end, we visualize the training samples that have the highest positive IS score.

### G.2 EXTENDING HARMFUL DATA REMOVAL TO REAL-WORLD NOISY DATASET

We use *ANIMAL-10N dataset* (Song et al., 2019) for our harmful data removal tasks. It's a real world noisy dataset, containing five pairs of "confusing" animals: (cat, lynx), (jaguar, cheetah), (wolf, coyote), (chimpanzee, orangutan), (hamster, guinea pig), where two animals in each pair look very similar. Overall, the proportion of incorrect labels was 6.44%.

The harmful removal task on ANIMAL-10N is presented in table 3. We observe that when a portion of harmful data is removed, the ECIF method significantly outperforms random removal, particularly when the removal proportion is small. Specifically, when less than 40% of the data is removed, ECIF achieves an accuracy improvement of over 10% compared to random removal. This demonstrates the capability of ECIF to accurately identify harmful samples, thereby substantially enhancing the model's performance.

To provide a method as the reference, we adopt CLIPScore (Hessel et al., 2021), a basic data evaluation method, as the baseline for MLLM. This method is model-independent and is limited to

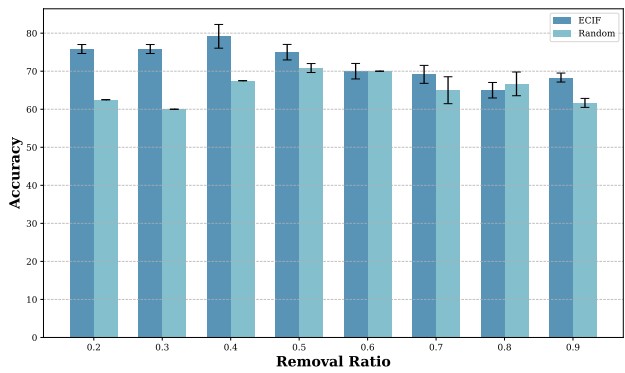

Figure 3: Harmful Data Removal on ANIMAL-10N

evaluating data quality rather than assessing the contribution of the data to the model. In the task

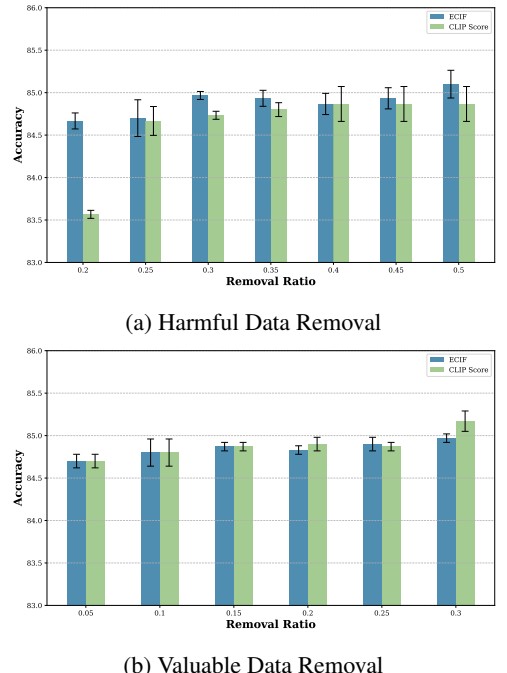

(a) Harmful Data Removal

(b) Valuable Data Removal

Figure 4: Comparison between different methods for data removal on Food101

of harmful data removal, the ECIF method demonstrates significantly better performance compared to the CLIP Score. For valuable data removal, ECIF performs slightly better than the CLIP Score. This superiority is primarily attributed to ECIF's ability to attribute data based on the relationship between the model and the data, whereas CLIP Score is solely used to evaluate data quality without considering the model's involvement.

## G.3 EVALUATING MULTIPLE SAMPLES

To comprehensively evaluate the data removal capabilities of ECIF in various scenarios, we conducted experiments on the performance when multiple samples need to be removed. Specifically, we consider the different ratios of samples (1-7%) for removal. As shown in Figure 5, we can see that the difference in precision between these two methods is very small (less than 1.5%) in most cases, except for the case of 2% for Food101. These results show the utility of ECIF compared to the ground truth. Note

that in Table 2, we have shown that the speed of ECIF is more than twice that of retraining. Thus, ECIF is an editing method that achieves a trade-off between speed and effectiveness.

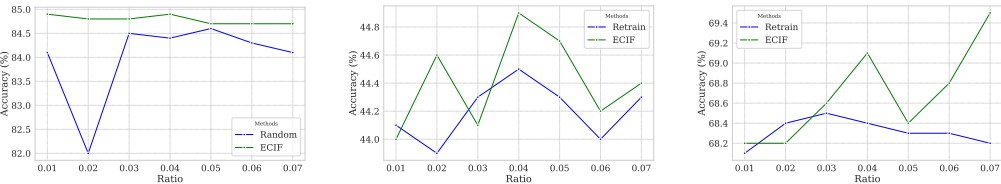

Figure 5: Impact of Remove Ratio on Food101, DTD and Flower102 datasets.

## G.4 ADDITIONAL VISUALIZATION OF MISPREDICTION TRACE BACK

We demonstrate our additional visualization results of the mispredicted data tracing in Table 4-6 and Figure 8 -10.

## G.5 ADDITIONAL VISUALIZATION OF MISALIGNMENT DATA DETECTION

We demonstrate our additional results of the Visualization of the misalignment data detection in Figure 6 - 7.

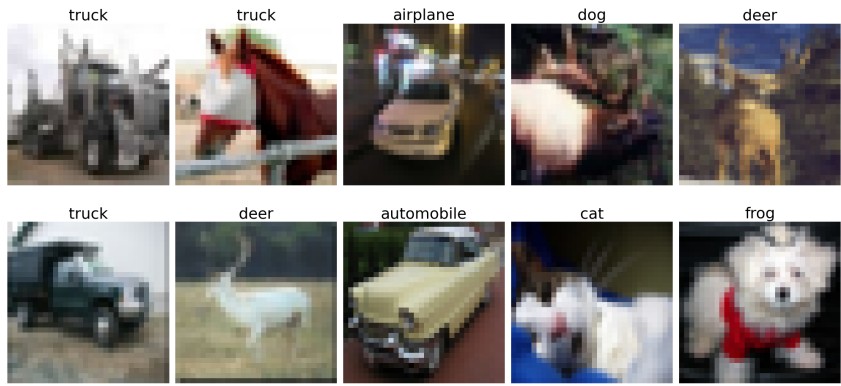

Figure 6: Top-10 misaligned sample pairs in the 20% mislabeled training data.

## G.6 LIMITATION

Our proposed data value evaluation method assesses the importance of a data point by performing leave-one-out retraining and observing changes in model performance. We use ECIF to efficiently estimate this performance change, thus enabling rapid data value assessment. Essentially, the ECIF approach approximates the results that would be obtained by actual leave-one-out retraining. However, experimental results demonstrate that this approximation is highly effective in real-world applications.

## G.7 EFFECT OF REMOVAL RATIO ON DATA FILTERING PERFORMANCE

We evaluated the effectiveness of our method through systematic data removal experiments on the Animal10 dataset. As illustrated in Figure 11(a), in the harmful data removal setting, ECIF demonstrates a stable performance improvement and consistently outperforms baseline methods (such as CLIP Score, D2 Pruning and Image Filter) as the removal ratio increases. This suggests that ECIF effectively isolates detrimental samples. Conversely, the useful data removal experiment in Figure 11(b) serves as a strong counter-validation. When the data deemed most valuable by ECIF is removed, the model exhibits the most significant accuracy drop compared to other methods.

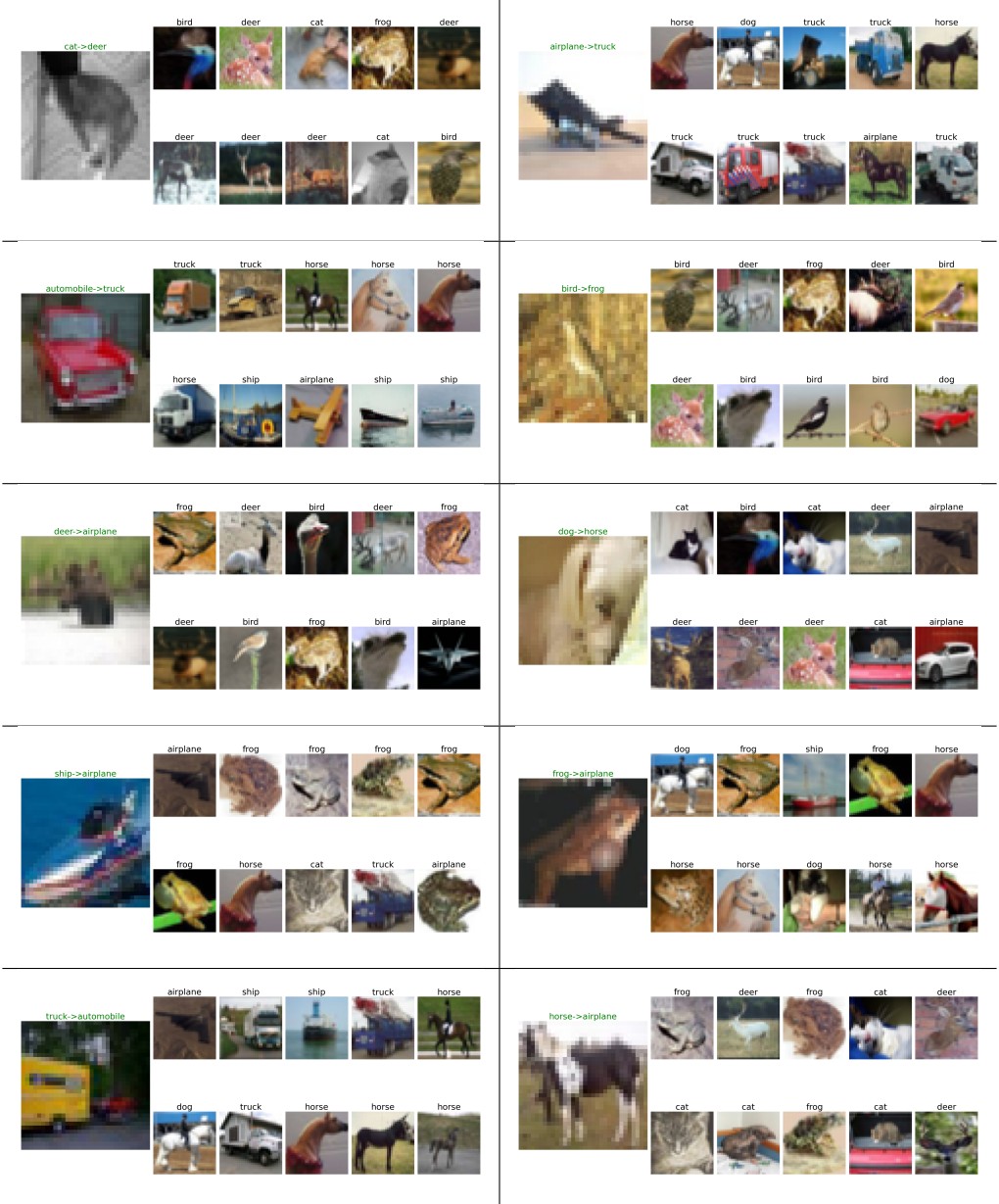

Table 4: Top-10 related test data tracing of mispredicted data on cifar-10 dataset with 10% noise data.

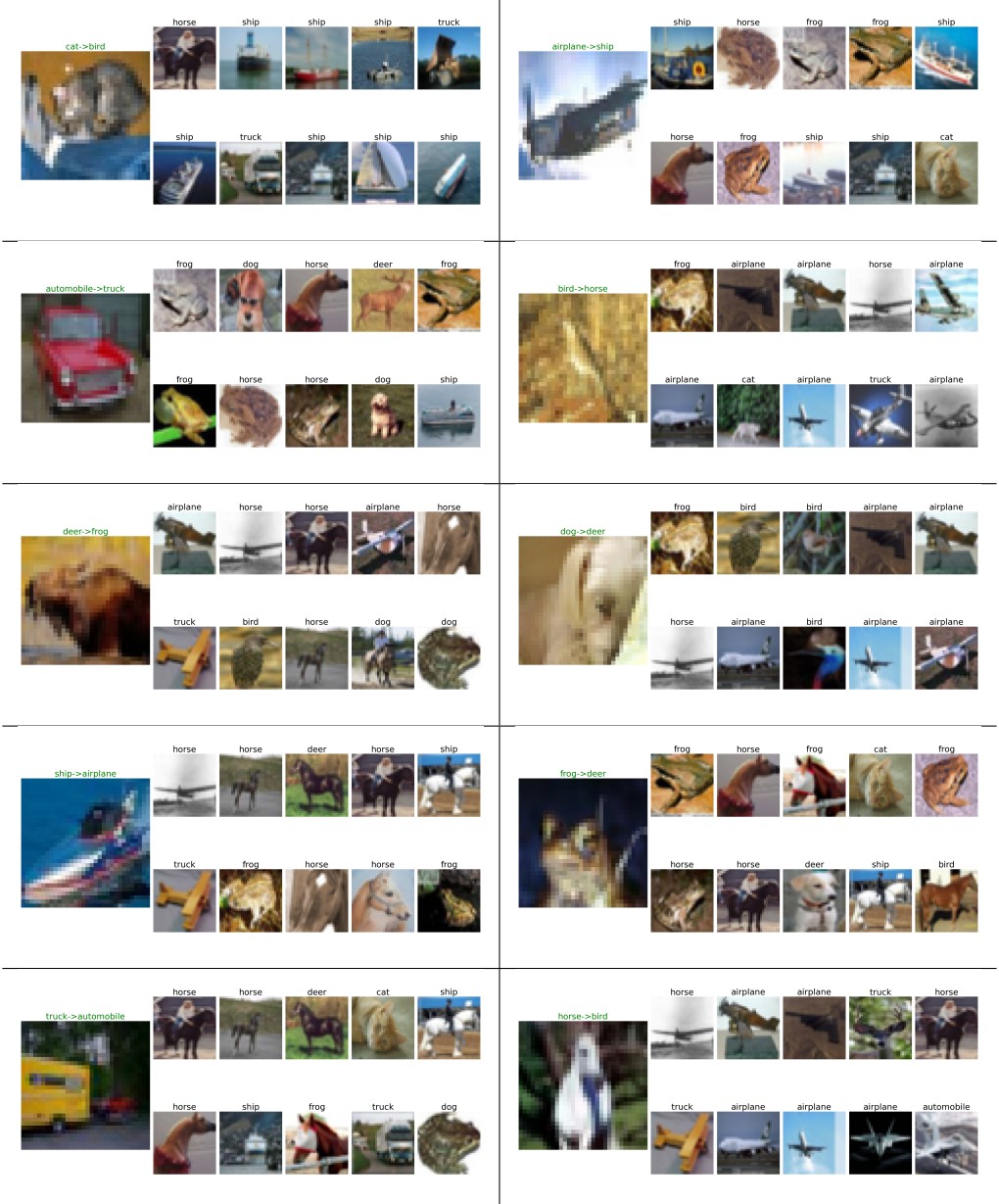

Table 5: Top-10 related test data tracing of mispredicted data on cifar-10 dataset with 20% noise data.

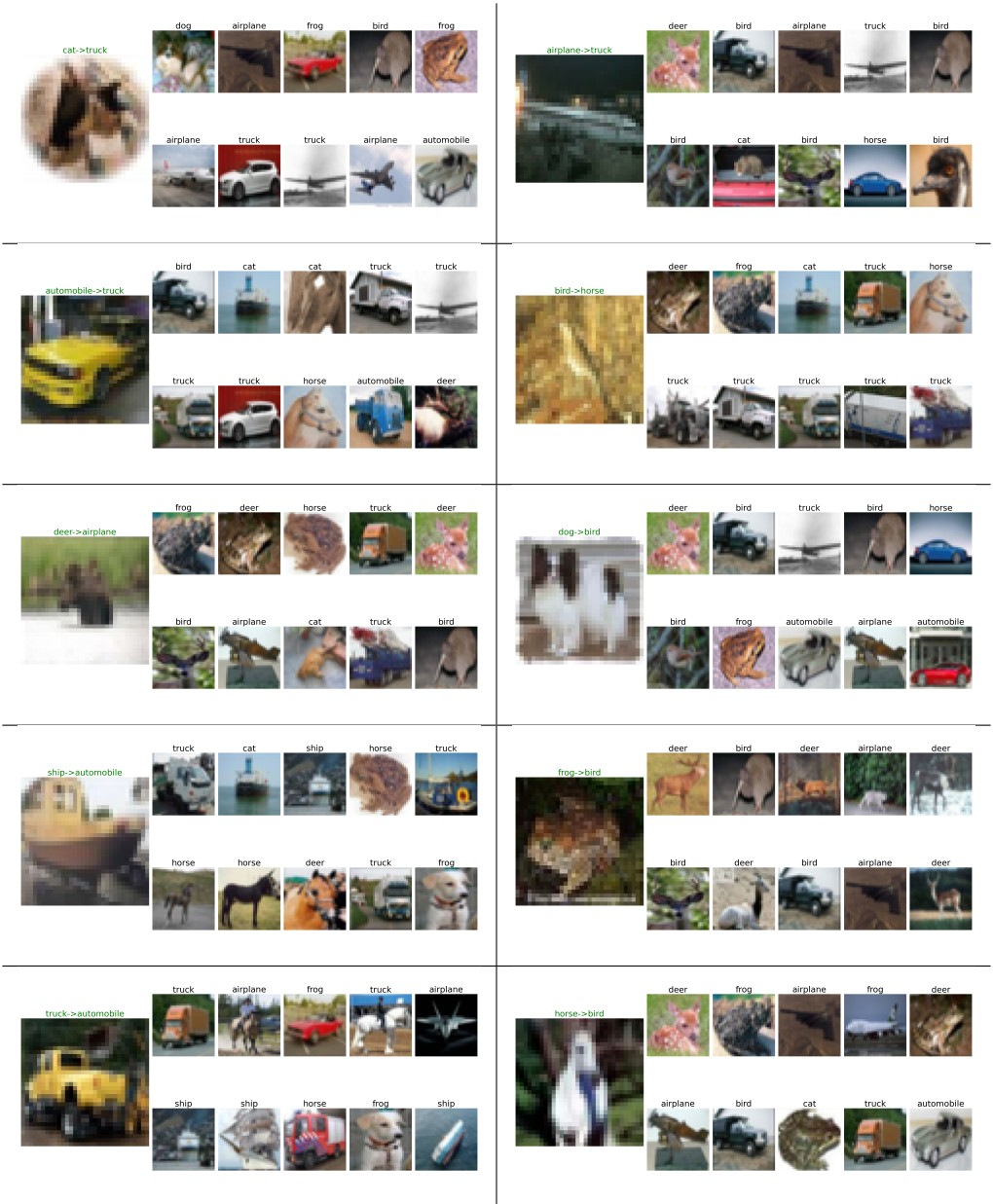

Table 6: Top-10 related test data tracing of mispredicted data on cifar-10 dataset with 30% noise data.

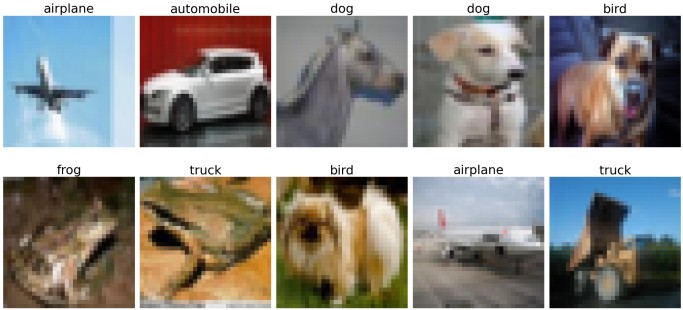

Figure 7: Visualization results for misalignment detection. 30% of the training samples were mislabeled. The figure shows the training samples that have the top-10 highest IS scores on the cifar-10 test set.

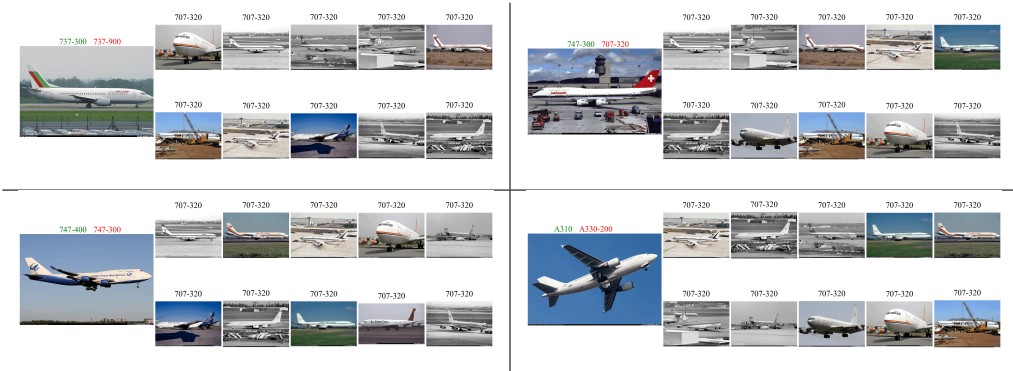

Figure 8: Top-10 related test data tracing of mispredicted data on FGVC-Aircraft with 30% noise data.

This sharp performance degradation confirms that ECIF accurately locates the critical samples that contribute most to the model's training.

### G.8 VISUAL COMPARISON OF SAMPLE SELECTION: ECIF VS. CLIP SCORE

To intuitively understand the selection mechanism, we visualized the most valuable samples ranked by ECIF and CLIP Score for specific target images. As shown in Figure 12, the comparison reveals distinct behaviors in handling the inherent noise of the Animal10N dataset. The CLIP Score tends to rely on superficial visual traits, frequently retrieving "hard negatives"—such as selecting a Coyote for a Wolf target—that look similar but belong to different classes. In contrast, ECIF effectively filters out these deceptive lookalikes and identifies semantically consistent samples. This suggests that our method moves beyond mere visual alignment to capture the true data value, avoiding the confusion that often arises from the high inter-class similarity in real-world noisy data.

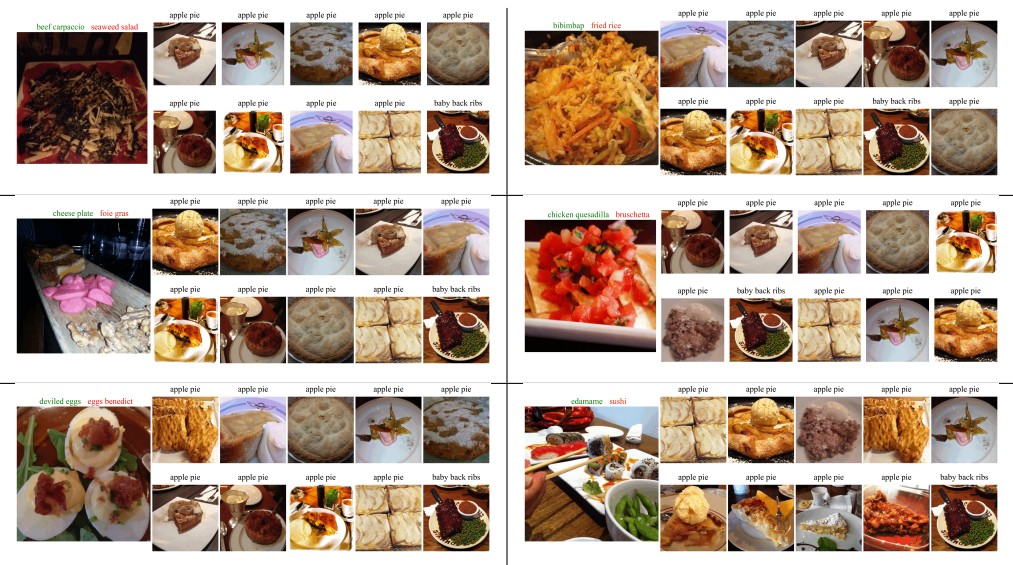

Figure 9: Top-10 related test data tracing of mispredicted data on Food-101 with 30% noise data.

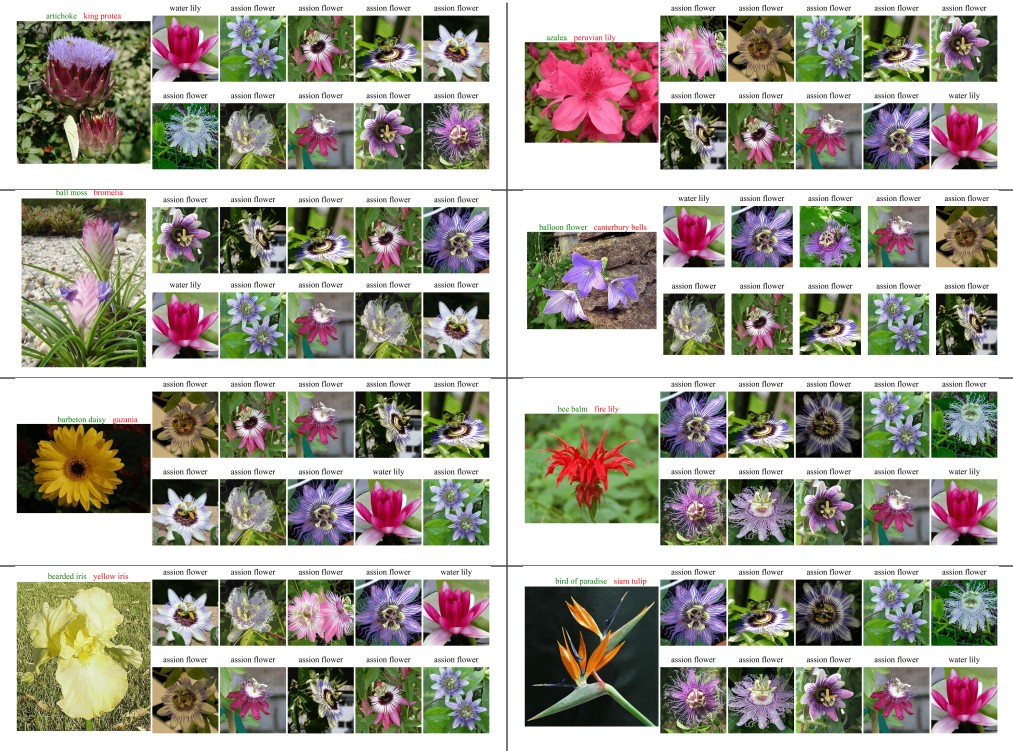

Figure 10: Top-10 related test data tracing of mispredicted data on Flowers-102 with 30% noise data.

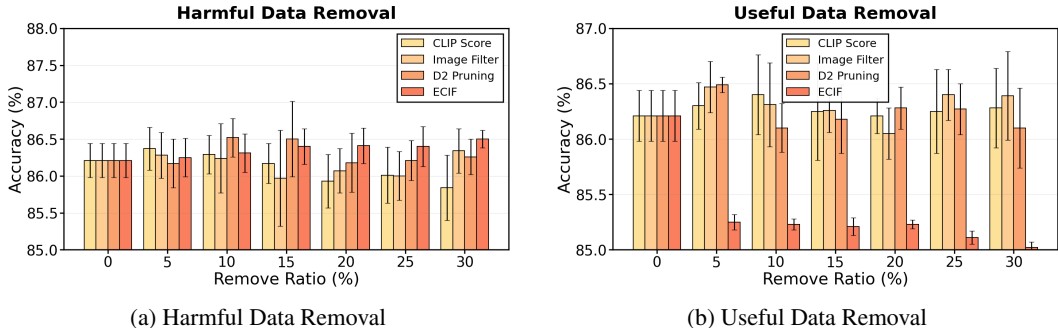

(a) Harmful Data Removal          (b) Useful Data Removal

Figure 11: Impact of different removal ratios on model performance.

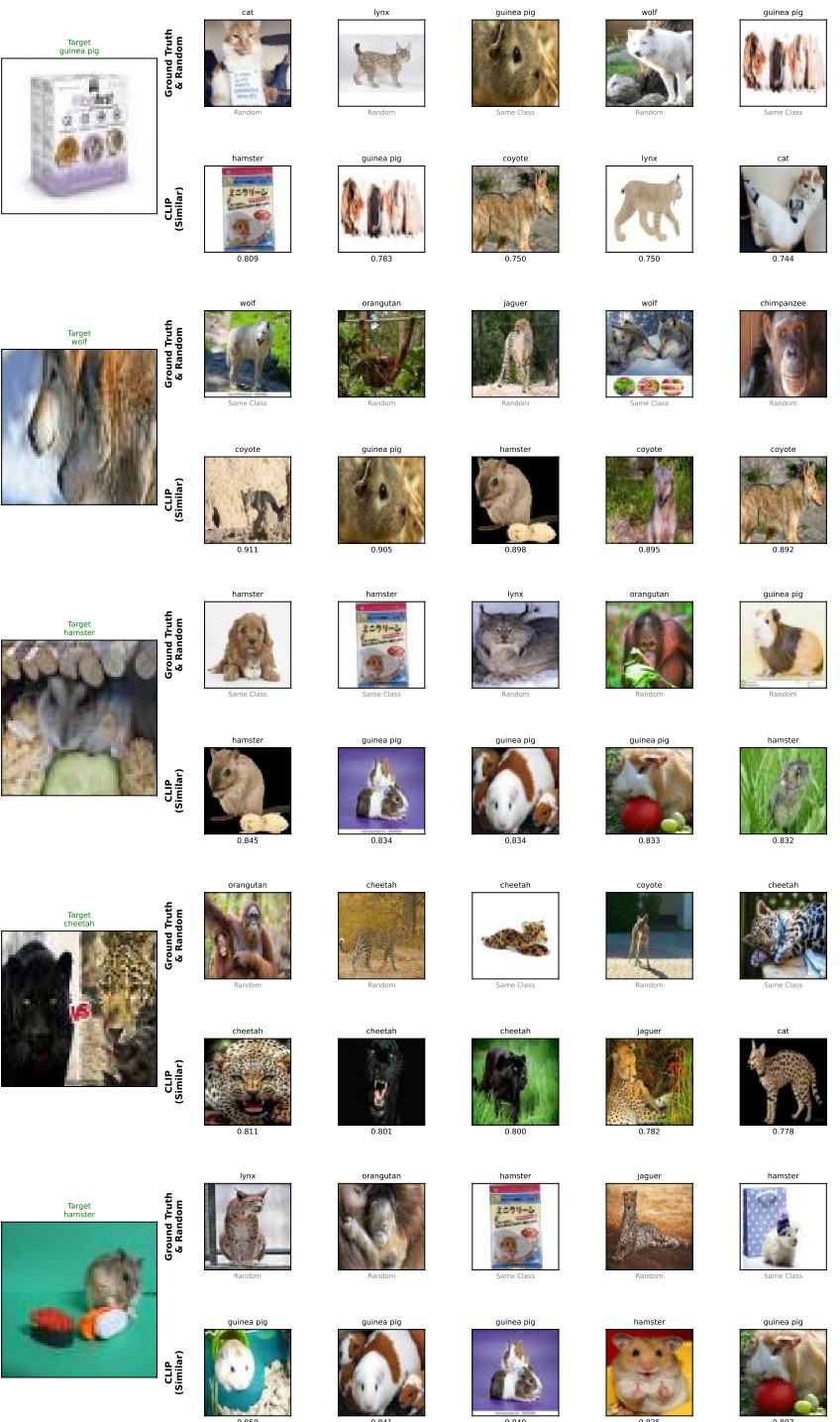

Figure 12: Visual comparison of selected most valuable samples using different metrics.

