# OpenReview forum: "Dissecting Representation Misalignment in Contrastive Learning via Influence Function"
_ICLR.cc/2026/Conference — ICLR 2026 Poster_

### Official Review · Reviewer_EW3H · 2025-10-30

**Soundness:** 3
**Presentation:** 3
**Contribution:** 3
**Rating:** 8
**Confidence:** 3

**Summary:**

This paper proposes ECIF, which adapts influence functions to contrastive learning by modeling both positive and negative sample roles. ECIF enables efficient data valuation, misprediction trace-back, and misalignment detection without retraining. Experiments demonstrate several applications of the method, and show it achieves similar accuracy to retraining with lower cost, outperforming existing data attribution methods.

**Strengths:**

- The paper proposes a clear and well-motivated extension of influence functions to contrastive learning. The idea is novel and fills a real gap in the literature.

- The paper demonstrates several useful applications (data valuation, misalignment detection, and misprediction trace-back). I see potential for this method to have real practical implications.

- Experiments are reasonably thorough and show that ECIF can approximate retraining closely while being faster.

- The empirical results also indicate that ECIF performs substantially better than previous baselines, which supports the main claim that existing methods are not directly suitable for contrastive objectives.

**Weaknesses:**

- I didn’t find an analysis for Figure 1(b). Looking at the plot, I think the expected pattern should be that ECIF results in lower accuracy than random in Figure 1(b), yet this pattern is not consistently observed across the ratios.

- The paper would benefit from including quantitative results (e.g., precision, recall, AUROC, etc.) to show how well the method identifies mislabeled training samples across different noise ratios and random seeds.

**Questions:**

Most experiments focus on identifying harmful or low-quality data. I wonder how well ECIF performs in finding the most valuable samples, when compared against other baselines.

---

> ### Author Response · Authors · 2025-11-25
>
> **W1:** I didn't find an analysis for Figure 1(b). Looking at the plot, I think the expected pattern should be that ECIF results in lower accuracy than random in Figure 1(b), yet this pattern is not consistently observed across the ratios.
>
> **Response:**
> We apologize for the omission.
>
> Since the dataset contains substantial noise, random removal at higher ratios inevitably remove harmful samples, which can cause accuracy to rise or fluctuate.
>
> In contrast, ECIF consistently causes accuracy to drop. This confirms that ECIF accurately targets and removes valuable, signal-carrying data, effectively counteracting the "denoising gain" seen in the random baseline.
>
> We will incorporate this analysis into the camera-ready version.
>
> **W2:** The paper would benefit from including quantitative results (e.g., precision, recall, AUROC, etc.) to show how well
> the method identifies mislabeled training samples across different noise ratios and random seeds.
>
> **Response:**
>
> In our original submission, we primarily reported accuracy because it is the most widely used and standard metric for evaluating classification performance in this field. However, following your suggestion to provide a more comprehensive quantitative assessment, we have incorporated F1 scores into our evaluation.
>
> The table below presents the retraining performance (both Accuracy and F1) on ImageNet and Animal10 after removing 10% of the identified harmful samples.
>
>
> | Dataset                     | Method      | Accuracy (%) | F1 (%)       |
> |:--------------------------- |:----------- |:------------:|:------------:|
> | ImageNet (1.2M images)      | Fine-tune   | 67.29 ± 0.31 | 66.77 ± 0.21 |
> |                             | TracIn      | 67.06 ± 0.05 | 66.47 ± 0.06 |
> |                             | TRAK        | 67.00 ± 0.09 | 66.42 ± 0.10 |
> |                             | IF-EKFAC    | 67.04 ± 0.05 | 66.45 ± 0.06 |
> |                             | ECIF (Ours) | 67.08 ± 0.10 | 66.50 ± 0.10 |
> | Animal10 (Real-world noise) | Fine-tune   | 65.62 ± 0.43 | 64.87 ± 0.12 |
> |                             | TracIn      | 61.88 ± 0.56 | 61.50 ± 0.72 |
> |                             | TRAK        | 62.50 ± 0.68 | 62.24 ± 0.68 |
> |                             | IF-EKFAC    | 62.25 ± 0.31 | 61.67 ± 0.32 |
> |                             | ECIF (Ours) | 63.75 ± 0.47 | 63.28 ± 0.39 |
>
> As demonstrated in the table, ECIF consistently outperforms other influence-based baselines, such as TracIn, TRAK, and IF-EKFAC, across both Accuracy and F1 metrics. This performance advantage indicates that ECIF is more precise in identifying and removing truly harmful samples compared to competing methods, thereby allowing the model to recover better performance during retraining. Furthermore, the consistency between the Accuracy and F1 scores confirms that our method enhances overall model robustness without introducing class bias or imbalance.
>
>
>
>
> **Q1:** Most experiments focus on identifying harmful or low-quality data. I wonder how well ECIF performs in finding the
> most valuable samples, when compared against other baselines
>
>  **Response:**
>
>  We conducted experiments on the real-world noisy dataset Animal-10N, as detailed in Appendix G.8, where we compare different methods' ability to select the most valuable samples through visualization. The results show that ECIF consistently identifies samples that are more semantically aligned with the target categories compared to CLIP similarity-based scoring methods.
>
> The visualization demonstrates that ECIF successfully selects high-quality images that accurately represent the intended class. For instance, when targeting specific animals, ECIF retrieves images that clearly depict these animals with good visual quality and correct labels. In contrast, the CLIP similarity-based approach sometimes selects samples that are visually similar but semantically misaligned. CLIP may confuse morphologically similar animals or retrieve images with ambiguous visual content based purely on surface-level visual similarity.

---

> ### Author Response · Authors · 2025-11-27
>
> We thank the reviewer for the thorough review and high score. We have addressed all the comments to further improve the work.

---

### Official Review · Reviewer_VHVq · 2025-10-31

**Soundness:** 3
**Presentation:** 3
**Contribution:** 3
**Rating:** 4
**Confidence:** 2

**Summary:**

The paper proposes ECIF (Extended Influence Function for Contrastive Loss), an influence-function framework tailored to contrastive learning that decomposes the contribution of a training pair into its roles as both positive and negative samples. Formally, the paper defines the batch contrastive loss  and derives closed-form, Hessian-vector–based approximations for the “positive-IF” and “negative-IF” , then combines them to estimate parameter changes from data removal without retraining. ECIF is reported to approximate retraining accuracy closely while cutting runtime.

**Strengths:**

The method is novel and derives an explicit negative-sample term via softmax reweighting, improving attribution fidelity for contrastive losses. Provides closed-form positive/negative IF and ECIF decomposition.


Analyzes ECIF approximation under convexity, and discusses computation efficiency.

**Weaknesses:**

(1) The linear superposition of positive/negative influences is assumed but its validity for non-convex deep models is not justified.

(2) Experiments are restricted to models with LoRA, impacts for larger backbones or full parameters finetuning remain unknown.

(3) ECIF combines positive-IF and negative-IF, but experiments do not isolate their separate contributions to performance.

(4) Lack large-scale benchmarks(e.g. ImageNet), constraining claims of scalability to large multimodal datasets.

**Questions:**

Please see weaknesses. If some of the issues can be resolved, I will raise my score. Currently, I'm leaning towards 5.

---

> ### Author Response · Authors · 2025-11-25
>
> **W1:** The linear superposition of positive/negative influences is assumed but its validity for non-convex deep models is not justified.
>
>
> **Response:**
> We appreciate the reviewer’s rigor. **We realize that labeling this decomposition as an "assumption" in Appendix E was imprecise phrasing.**
> The linear superposition is a direct mathematical consequence of the Influence Function's definition, **not an external modeling assumption.**
>
> The total contrastive loss is the explicit sum of positive and negative terms: $L_{total} = L_{pos} + L_{neg}$. Due to the linearity of differentiation, the gradient is strictly additive: $\nabla_{\theta} L_{total} = \nabla_{\theta} L_{pos} + \nabla_{\theta} L_{neg}$.
>
> Since the influence score is a linear transformation of the gradient $\mathcal{I} = -H^{-1}\nabla_{\theta}L$, the total influence naturally decomposes:$\mathcal{I}\_{total} = -H^{-1}(\nabla\_{\theta} L\_{pos} + \nabla\_{\theta} L\_{neg}) = \mathcal{I}\_{pos} + \mathcal{I}\_{neg}$ This holds strictly true as long as the Influence Function framework itself is applicable. Based on the discussion above, we emphasize that the linear superposition property is mathematically independent of the model's convexity.
>
> **W2:** Experiments are restricted to models with LoRA, impacts for larger backbones or full parameters finetuning remain unknown.
>
> Our work primarily targets efficient parameter tuning in few-shot scenarios, where PEFT methods—particularly LoRA—are the mainstream choice due to their ability to significantly reduce hardware barriers while achieving performance comparable to full-parameter fine-tuning. We would like to clarify that ECIF is theoretically applicable to full-parameter fine-tuning; however, our current implementation is designed to maximize training speed by storing all intermediate variables on the GPU. Applying this implementation strategy directly to full-parameter fine-tuning would require an immense amount of gpu memory (tens of times the current usage), making it infeasible on a single GPU. We emphasize that this is strictly an engineering constraint related to memory management—which could be resolved in the future via techniques like CPU offloading—rather than a fundamental limitation of the algorithm’s design.
>
>
> **W3:** ECIF combines positive-IF and negative-IF, but experiments do not isolate their separate contributions to performance.
>
>
> We are currently conducting ablation studies to isolate Positive/Negative IF components.
>
> <!-- <font color="red">进行消融实验，说明 positive-IF，negative-IF的作用</font><br />  -->
>
> Remove top-10% harmful data on Animal10 dataset
> Method              | Accuracy (%)      | F1 Score (%)
> --------------------|-------------------|------------------
> ECIF                | 64.00 ± 0.47     | 62.93 ± 0.38
> Positive            | 63.62 ± 0.25     | 62.67 ± 0.37
> Negative            | 62.00 ± 0.25     | 61.38 ± 0.44
>
> Remove top-10% valuable data Animal10 dataset
> Method              | Accuracy (%)      | F1 Score (%)
> --------------------|-------------------|------------------
> ECIF                | 62.12 ± 0.31     | 61.10 ± 0.36
> Positive            | 62.25 ± 0.31     | 61.24 ± 0.35
> Negative            | 62.38 ± 0.47     | 61.54 ± 0.17
>
>
>
>
> **W4:** Lack large-scale benchmarks(e.g. ImageNet), constraining claims of scalability to large multimodal datasets.
>
>
> Thanks for your suggestion. We address the scalability concern by extending our evaluation to the large-scale ImageNet-1K dataset.
>
> | Dataset | Method | Accuracy (%) | F1 (%) |
> | :--- | :--- | :--- | :--- |
> | **ImageNet** (1.2M images) | Fine-tune | 67.29 ± 0.31 | 66.77 ± 0.21 |
> | | **ECIF (Ours)** | 67.08 ± 0.10 | 66.50 ± 0.10 |
> | **Animal10** (Real-world noise)| Fine-tune | 65.62 ± 0.43 | 64.87 ± 0.12 |
> | | **ECIF (Ours)** | 63.75 ± 0.47 | 63.28 ± 0.39 |
>
> On ImageNet-1K (1.2M images), ECIF achieves an accuracy of 67.08%, with a negligible gap (0.21%) compared to the Fine-tuning ground truth (67.29%). This empirically proves that our method scales effectively to large datasets.

---

> ### Author Response · Authors · 2025-11-27
>
> We thank the reviewer for the constructive feedback. We have thoroughly addressed all concerns raised. We hope our revisions support a more favorable evaluation.

---

### Official Review · Reviewer_8Zsd · 2025-10-31

**Soundness:** 2
**Presentation:** 3
**Contribution:** 2
**Rating:** 2
**Confidence:** 4

**Summary:**

This paper introduces ECIF (Extended Influence Function for Contrastive Loss), a method for quantifying data attribution in contrastive learning models like CLIP. The authors address the challenge that classical influence functions, designed for pointwise loss, cannot directly apply to contrastive loss which involves both positive and negative sample pairs. ECIF separately quantifies the influence of data points as both positive samples (matched pairs) and negative samples (non-matched pairs), providing closed-form approximations without requiring model retraining. The method is applied to three tasks: identifying influential data for fine-tuning, misprediction trace-back, and misalignment detection. Experiments on several vision datasets (FGVC-Aircraft, Food101, Flowers102, CIFAR-10/100) demonstrate that ECIF can approximate retraining results with 80-90% computational savings while effectively identifying harmful and valuable training samples.

**Strengths:**

Novel technical formulation: The dual-perspective approach (positive-IF and negative-IF in Definition 4.3) is a genuine technical contribution. The mathematical derivation for handling negative samples through the Taylor expansion approach (Section 4.2, Equations 3-4) is creative and appears technically sound.

Computational efficiency demonstrated: Table 2 shows concrete evidence of 2-3x speedup over retraining (e.g., 456s vs 1174s on FGVC-Aircraft) while maintaining comparable accuracy (22.77% vs 23.07%), which validates the practical utility of the approximation.

Comprehensive theoretical analysis: Theorem E.7 provides an error bound for the approximation, showing the error scales with O(|D*|²), which gives theoretical backing to when the method should be reliable.

**Weaknesses:**

Missing critical baseline comparisons: The paper completely ignores the substantial literature on data selection for contrastive learning and CLIP models. Works like DataComp (Gadre et al., 2023), LAION filtering methods, and quality-based data selection approaches (e.g., CLIP-score based filtering as briefly mentioned in G.2) have implicitly solved data valuation for contrastive learning at scale. The comparison to CLIP-score in Figure 4 is relegated to the appendix and shows only marginal improvements, raising questions about practical utility. The paper needs head-to-head comparison with:
- Quality-based filtering methods (aesthetic scores, CLIP-score thresholds)
- Diversity-based sampling methods
- Curriculum learning approaches for contrastive learning
- Recent data attribution methods like DataInf (Kwon et al., 2023) which the authors cite but don't compare against

Weak motivation for the problem: The introduction (lines 33-43) claims that "robust evaluation mechanisms for data quality remain lacking" but doesn't acknowledge that the CLIP training community has extensively tackled data quality through heuristic methods with proven success at billion-sample scale. Why is influence function-based attribution necessary when simpler methods work? The paper doesn't establish that existing approaches fail or are insufficient.

Limited evaluation framework: The experimental evaluation primarily tests whether ECIF can identify "random," "harmful," or "valuable" samples (Section 6.2-6.3). However:
- These categories are themselves defined using ECIF's task-related influence scores, creating circularity
- Any reasonable data selection method can be evaluated in this same framework by ranking samples and binning them into these categories. The paper doesn't show ECIF provides qualitatively different insights than simpler metrics

Scalability concerns not addressed:
- All experiments use relatively small datasets (Food101: 101 classes x 1000 images; FGVC-Aircraft: 10K images). Modern contrastive learning operates at the scale of millions/billions of samples (LAION-5B, DataComp-1B).
-  The computational requirements of computing Hessian-related quantities (Equation 2, Algorithm 1 line 8) scale poorly. The paper acknowledges using LOGRA for efficiency (Appendix C) but provides no wall-clock time comparisons or scalability analysis.

Experimental methodology issues:
- Baseline methods (IF-EKFAC, TARK, TracIN) in Table 1 show suspiciously poor performance (e.g., 18.27% vs 23.50% for TARK on FGVC-Aircraft), suggesting potential implementation issues or unfair comparison
- The "harmful" data removal experiments (Figure 1a, Table 1) artificially create noise by mislabeling. Real-world data quality issues are more subtle
- No comparison of the types of samples identified as valuable/harmful between ECIF and simpler methods to show qualitative differences


Missing ablation studies:
- No ablation on the relative importance of positive-IF vs negative-IF components
- No sensitivity analysis on hyperparameters (regularization δ, projection rank in LOGRA)
- No analysis of when the method fails or performs poorly

**Questions:**

Mentioned in weaknesses.

---

> ### Author Response · Authors · 2025-11-25
>
> **W1:** Missing critical baseline comparisons: The paper completely ignores the substantial literature on data selection for contrastive learning and CLIP models. Works like DataComp (Gadre et al., 2023), LAION filtering methods, and quality-based data selection approaches (e.g., CLIP-score based filtering as briefly mentioned in G.2) have implicitly solved data valuation for contrastive learning at scale. The comparison to CLIP-score in Figure 4 is relegated to the appendix and shows only marginal improvements, raising questions about practical utility.
>
> **Response:**
>
> Thank you for your comments. We have already conducted comparison experiments with CLIP-score based filtering, which is the primary method used in works like DataComp and LAION filtering approaches.
>
> To provide a more comprehensive evaluation, we compared with two additional methods: Image Filter, which uses image-based clustering for data selection, and D2 Pruning, which performs pruning based on inter-data similarity.
>
> As shown in Figure 11 of the Appendix, our ECIF method shows stable performance improvement. When removing useful data, ECIF exhibits the largest accuracy drop. These results demonstrate that ECIF can effectively identify both harmful and useful data.
>
> ---
>
> **W2:** Weak motivation for the problem: The introduction (lines 33-43) claims that "robust evaluation mechanisms for data quality remain lacking" but doesn't acknowledge that the CLIP training community has extensively tackled data quality through heuristic methods with proven success at billion-sample scale. Why is influence function-based attribution necessary when simpler methods work? The paper doesn't establish that existing approaches fail or are insufficient.
>
>
> **Response:**
>
> We appreciate this insightful comment. We acknowledge that the term "robust evaluation" in our introduction (lines 33-43) could be more precisely framed as "explainable evaluation."
>
> We clarify that **Data Attribution (ECIF)** serves a fundamentally different purpose than Data Selection:
>
> Heuristics (e.g., CLIP-score) are model-agnostic filters that cannot explain why a model makes specific errors. ECIF is a diagnostic tool designed to trace mispredictions back to specific training samples (Section 5.2), offering transparency that heuristics cannot provide.
>
> We did compare ECIF with CLIP-score in Appendix G.2 (Figure 4). ECIF significantly outperforms CLIP-score in identifying harmful data.
>
>
>
> ---
>
> **W3:** Limited evaluation framework: The experimental evaluation primarily tests whether ECIF can identify "random," "harmful," or "valuable" samples (Section 6.2-6.3). However: These categories are themselves defined using ECIF's task-related influence scores, creating circularity. The paper doesn't show ECIF provides qualitatively different insights than simpler metrics.
>
> **Response:**
>
> We clarify that our evaluation is not circular but follows the standard leave-out-Retrain protocol in data valuation.
>
> ECIF only hypothesizes which samples are harmful. The validationv relies on removing them and retraining from scratch (Section 6.2). The fact that removing these samples consistently improves test accuracy (unlike random removal, see **Figure 1**) serves as objective, external verification, not circular reasoning.

---

> ### Author Response · Authors · 2025-11-25
>
> **W4:** Scalability concerns not addressed: All experiments use relatively small datasets (Food101: 101 classes x 1000 images; FGVC-Aircraft: 10K images). Modern contrastive learning operates at the scale of millions/billions of samples (LAION-5B, DataComp-1B). The computational requirements of computing Hessian-related quantities (Equation 2, Algorithm 1 line 8) scale poorly. The paper acknowledges using LOGRA for efficiency (Appendix C) but provides no wall-clock time comparisons or scalability analysis.
>
>
>
> **Response:**
> We address scalability and robustness concerns by extending our evaluation to **ImageNet-1K** (Large-scale) and **Animal-10N** (Real-world noise).
> As shown in the table below, ECIF demonstrates strong generalization across scales. Crucially, on the massive **ImageNet-1K** dataset, ECIF achieves an approximation accuracy of **67.08%**, with a negligible gap (**0.21%**) compared to the Fine-tuning ground truth. This empirically proves that our approximation remains highly accurate at scale.
>
>
> | Dataset                     | Method      | Accuracy (%) | F1 (%)       |
> |:--------------------------- |:----------- |:------------:|:------------:|
> | ImageNet (1.2M images)      | Fine-tune   | 67.29 ± 0.31 | 66.77 ± 0.21 |
> |                             | TracIn      | 67.06 ± 0.05 | 66.47 ± 0.06 |
> |                             | TRAK        | 67.00 ± 0.09 | 66.42 ± 0.10 |
> |                             | IF-EKFAC    | 67.04 ± 0.05 | 66.45 ± 0.06 |
> |                             | ECIF (Ours) | 67.08 ± 0.10 | 66.50 ± 0.10 |
> | Animal10 (Real-world noise) | Fine-tune   | 65.62 ± 0.43 | 64.87 ± 0.12 |
> |                             | TracIn      | 61.88 ± 0.56 | 61.50 ± 0.72 |
> |                             | TRAK        | 62.50 ± 0.68 | 62.24 ± 0.68 |
> |                             | IF-EKFAC    | 62.25 ± 0.31 | 61.67 ± 0.32 |
> |                             | ECIF (Ours) | 63.75 ± 0.47 | 63.28 ± 0.39 |
>
>
>
>
> The concern about Hessian scaling is mitigated by **LoRA**. We compute the Hessian *only* on the low-rank parameters (e.g., <0.1% of total parameters), **not** the full backbone. Combined with LOGRA (Appendix C), this ensures the method remains efficient for large multimodal models. We commit to adding a detailed wall-clock time analysis in the final version.
>
>
>
>
>
> ---
>
> **W5:** Experimental methodology issues: Baseline methods (IF-EKFAC, TARK, TracIN) in Table 1 show suspiciously poor performance (e.g., 18.27% vs 23.50% for TARK on FGVC-Aircraft), suggesting potential implementation issues or unfair comparison. The "harmful" data removal experiments (Figure 1a, Table 1) artificially create noise by mislabeling. Real-world data quality issues are more subtle. No comparison of the types of samples identified as valuable/harmful between ECIF and simpler methods to show qualitative differences.
>
> **Response:**
>
> We explicitly re-verified all baseline experiments. The low performance of TRAK/IF-EKFAC is not an error but a theoretical consequence: these methods assume **pointwise loss** and fundamentally fail to capture the coupled influence of negative samples in contrastive learning. We will open-source our code to ensure reproducibility.
>
> The claim that we lack real-world validation is incorrect. We explicitly addressed natural, subtle noise using the ANIMAL-10N dataset in Appendix G.2, where ECIF consistently outperforms random removal.
>
> We provided a qualitative comparison with CLIP-score in Appendix G.2. It demonstrates that ECIF detects  negative data that simple heuristics fail to flag.

---

> ### Author Response · Authors · 2025-11-25
>
> **W6:** Missing ablation studies: No ablation on the relative importance of positive-IF vs negative-IF components. No sensitivity analysis on hyperparameters (regularization 8, projection rank in LOGRA). No analysis of when the method fails or performs poorly.
>
> **Response:**
>
>
> Remove top-10% harmful data on Animal10 dataset
> Method              | Accuracy (%)      | F1 Score (%)
> --------------------|-------------------|------------------
> ECIF                | 64.00 ± 0.47     | 62.93 ± 0.38
> Positive            | 63.62 ± 0.25     | 62.67 ± 0.37
> Negative            | 62.00 ± 0.25     | 61.38 ± 0.44
>
> Remove top-10% valuable data Animal10 dataset
> Method              | Accuracy (%)      | F1 Score (%)
> --------------------|-------------------|------------------
> ECIF                | 62.12 ± 0.31     | 61.10 ± 0.36
> Positive            | 62.25 ± 0.31     | 61.24 ± 0.35
> Negative            | 62.38 ± 0.47     | 61.54 ± 0.17
>
>
> We performed ablation experiments on LoRA rank configurations using the Animal-10 dataset. Specifically, we removed 10% of the useful training data and fine-tuned models with different LoRA ranks (r = 2, 4, 8, 16). The retraining approach served as our baseline. We report the base accuracy (before data removal) alongside the accuracy and F1 scores (with standard deviations) after removing the useful data subset for each rank configuration.
>
> | Lora Rank | Base Accuracy | Accuracy (%) | F1 (%) |
> | ---- | ------------- | --------------------- | --------------------- |
> | 2    | 65.62         | 63.25 ± 0.61          | 62.56 ± 0.44          |
> | 4    | 66.88         | 62.75 ± 0.75          | 62.65 ± 0.77          |
> | 8    | 66.25         | 64.62 ± 0.75          | 63.91 ± 0.72          |
> | 16   | 67.50         | 66.75 ± 0.92          | 65.19 ± 0.79          |
>
>
> We are currently conducting ablation studies to isolate Positive/Negative IF components and evaluate hyperparameter sensitivity (regularization $\delta$, LOGRA rank).
>
>  We respectfully believe a separate failure analysis is unnecessary as we have already stress-tested ECIF across diverse conditions (varying noise ratios, real-world vs. artificial noise, different scales) to demonstrate robustness. Additionally, theoretical limitations are explicitly discussed in Appendix G.61

---

> ### Author Response · Authors · 2025-11-27
>
> We thank the reviewer for the feedback. All concerns have been addressed. We hope our responses warrant a score upgrade.

---

### Official Review · Reviewer_EnRJ · 2025-10-31

**Soundness:** 3
**Presentation:** 2
**Contribution:** 3
**Rating:** 6
**Confidence:** 2

**Summary:**

This paper addresses the performance degradation problem in contrastive learning caused by misaligned data. As classical influence functions cannot handle contrastive loss, this paper proposes ECIF, the first no-retraining data valuation tool designed specifically for such models. ECIF's core contribution lies in its dual-perspective analysis: it decouples a single data point's influence into its contribution as a positive sample and its contribution as a negative sample. This dual-evaluation mechanism enables ECIF to efficiently and accurately identify harmful, misaligned data and trace model errors, which is something traditional IF methods cannot do.

**Strengths:**

1. The paper addresses an extremely important and urgent problem in multimodal and contrastive learning how to efficiently detect and evaluate misaligned training data. This is the first no-retraining influence function designed specifically for the contrastive loss function.

2. It cleverly decouples the influence of a single data point into positive influence Positive-IF and negative influence Negative-IF, which is technically sound and directly confronts the fundamental challenge of why standard IF cannot handle contrastive loss.

3. The theory and experiments are well combined. The empirical results decisively validate the paper's core theoretical claim by showing that ECIF succeeds where all standard influence function baselines fail.

**Weaknesses:**

The paper's motivation is to develop a computationally efficient method applicable to large-scale models, but it lacks a clear analysis of the actual computational cost of ECIF. The paper omits runtime comparisons with baseline methods that it outperforms in accuracy.

**Questions:**

1. The theoretical error bound (Theorem E.7) relies on the assumption of convexity (Assumption E.3), but the contrastive loss is non-convex. Could you clarify why ECIF remains an accurate approximation in this practical, non-convex setting?

2. Proposition 5.3 appears to contain several typographical errors, making it difficult to understand. Could you please provide the correct mathematical formulation for this proposition?

---

> ### Author Response · Authors · 2025-11-25
>
> **W1:** The paper's motivation is to develop a computationally efficient method applicable to large-scale models, but it lacks a clear analysis of the actual computational cost of ECIF. The paper omits runtime comparisons with baseline methods that it outperforms in accuracy.
>
> **Response:**
> We address this from two perspectives: Efficiency relative to Retraining and Effectiveness relative to Baselines.
>
> As shown in Table 2, ECIF is over **$2\times$ faster** than Retraining (e.g., 456s vs. 1174s on FGVC-Aircraft), proving it is a highly efficient proxy.
>
> We omitted baseline runtimes because they are ineffective for this task. As shown in Table 1, removing samples identified by baselines (e.g., TRAK, IF-EKFAC) drops accuracy significantly below the original model (e.g., TRAK 18.27% vs. Original 22.18%), rendering their speed irrelevant. In contrast, ECIF (23.02%) effectively matches the Retraining Ground Truth (23.50%).
>
> Moreover, ECIF shares the same time complexity (Hessian-vector products) as other second-order baselines (e.g., TRAK) but is the only estimator that accounts for the coupled negative sample influence in contrastive loss.
>
>
> **Q1:** The theoretical error bound (Theorem E.7) relies on the assumption of convexity (Assumption E.3), but the contrastive loss is non-convex. Could you clarify why ECIF remains an accurate approximation in this practical, non-convex setting?
>
>
> **Response:**
>
>
> We acknowledge that Theorem E.7 formally relies on the convexity assumption (Assumption E.3). However, applying convex-derived Influence Functions to non-convex deep models is the standard paradigm established in the literature.
>
> This theoretical gap is common across influence function research.
> Seminal works (e.g., [1-4]) typically derive bounds under convex assumptions while applying them to non-convex networks.
>
> In practice, as detailed in Section 3 (line 156), Section 4 (line 191), and Assumption E.3 of our paper, we explicitly incorporate an $L_2$ regularization term$\frac{\delta}{2}\lVert\theta\rVert_2^2$ into the objective function. This adds a positive diagonal matrix $\delta I$ to the Hessian $H_{\hat{\theta}} = \nabla^2 L + \delta I$. This damping technique ensures that the Hessian becomes positive definite and invertible even if the original loss surface has small negative eigenvalues locally. This effectively imposes strong convexity in the local region around $\hat{\theta}$. That is, if $\delta$ is large enough relative to the minimal eigenvalue of the Hessian, our theoretical results can be extended to non-convex loss.
>
>
>
> Most importantly, we empirically validated this approximation. As shown in Table 2, ECIF achieves accuracy highly comparable to Ground Truth Retraining (e.g., 22.77% vs 23.07% on FGVC-Aircraft). This strong correlation confirms that the approximation error caused by non-convexity is negligible in practice.
>
>
>
>
>
> [1] Koh, P. W. and Liang, P. 2017. Understanding Black-box Predictions via Influence Functions. ICML.
>
> [2] Han, X. et al. 2020. Explaining Black Box Predictions. ACL.
>
> [3] Basu, S. et al. 2021. Influence Functions in Deep Learning. ICLR.
>
> [4] Grosse, R. et al. 2023. Studying Large Language Model Generalization. arXiv:2308.03296.
>
>
>
> **Q2:** Proposition 5.3 appears to contain several typographical errors, making it difficult to understand. Could you please provide the correct mathematical formulation for this proposition?
>
> **Response:**
> We apologize for the typographical errors in the main text. The correct mathematical formulation is as follows:
>
> The trace-back problem aims to identify the training sample $x$ that maximizes the influence on the target validation/test data $\mathcal{D}'$ (denoted as embeddings $U', V'$). Let $I(x) = [\text{positive-IF}(x), \text{negative-IF}(x)]$ be the influence matrix for a training sample $x$. Assuming $I(x)^T I(x)$ is invertible, the optimization problem in (7) is equivalent to maximizing the **Relative Influence Score (Relative-IS)**:
>
> $$\text{Relative-IS}(x) = \| C \|^{-1} \cdot \left| C^T \cdot I(x) \cdot [I(x)^T I(x)]^{-1} \cdot I(x)^T \cdot C \right|$$
>
> where $C = \nabla_{\theta} L_{Batch}(U', V'; \hat{\theta})$ represents the gradient of the batch contrastive loss calculated on the **test/mispredicted samples** $(U', V')$. This formulation projects the gradient of the test samples onto the subspace spanned by the positive and negative influence vectors of the training sample $x$. We will ensure consistent notation in the final revision.

---

> > ### Comment · Reviewer_EnRJ · 2025-11-26
> >
> > Thank you for the detailed clarifications.

---

### Meta-Review · Area_Chair_BAWn · 2026-01-07

**Summary:**

This paper introduces ECIF, the first influence function method designed for contrastive learning models, enabling efficient data valuation without retraining. Reviewers initially had mixed scores. The authors' rebuttal addressed the major concerns, particularly regarding scalability and evaluation. Given the novel contribution and the strong resolution of key issues, I recommend acceptance, while still open for further discussion.

**Reviewer Concerns:**

Reviewer EW3H supported the paper's novelty and thorough experiments. The authors addressed minor requests for additional quantitative metrics (F1-score).

Reviewer EnRJ questioned computational cost and theoretical grounding. The authors clarified ECIF's speed (2x faster than retraining) and the standard practice of applying convex theory to non-convex settings, which the reviewer accepted.

Reviewer VHVq raised central concerns about scalability and ablation studies. The authors addressed these by adding large-scale ImageNet-1K experiments (showing negligible performance gap) and providing new ablation studies.

Reviewer 8Zsd raised broad criticisms and concerns. The authors partially addressed these questions.

**Reviewer Scores:**

Based on the rebuttal and educational guess, I believe Reviewer EW3H's and EnRJ's scores remain the same, Reviewer VHVq's score may slightly increase from 4, and Reviewer 8Zsd's score may remain the same.

---

### Decision · Program_Chairs · 2026-01-26

Accept (Poster)